# Bias correction of Simulated Historical Daily Streamflow at Ungauged Locations by Using Independently Estimated Flow-Duration Curves

William H. Farmer[1], Thomas M. Over[2], and Julie E. Kiang[3]

[1]U.S. Geological Survey, Denver, Colorado, United States
[2]U.S. Geological Survey, Urbana, Illinois, United States
[3]U.S. Geological Survey, Reston, Virginia, United States

**Correspondence:** William H. Farmer (wfarmer@usgs.gov)

**Abstract.** In many simulations of historical daily streamflow distributional bias arising from the distributional properties of residuals has been noted. This bias often presents itself as an underestimation of high streamflow and an overestimation of low streamflow. Here, 1168 streamgages across the conterminous United States having at least 14 complete water years of daily data between October 01, 1980, and September 30, 2013, are used to explore a method for rescaling simulated streamflow to correct the distributional bias. Based on an existing approach that separates the simulated streamflow into components of temporal structure and magnitude, the temporal structure is converted to simulated nonexceedance probabilities and the magnitudes are rescaled using an independently estimated flow-duration curve (FDC) derived from regional regression. In this study, this method is applied to a pooled ordinary kriging simulation of daily streamflow coupled with FDCs estimated by regional regression on basin characteristics. The improvement in the representation of high and low streamflows is correlated with the accuracy and unbiasedness of the estimated FDC. The method is verified by using an idealized case; however, with the introduction of regionally regressed FDCs developed for this study, the method is only useful overall for the upper tails, which are more accurately and unbiasedly estimated than the lower tails. It remains for future work to determine how accurate the estimated FDCs need to be to be useful for bias correction without unduly reducing accuracy. In addition to its potential efficacy for distributional bias correction, this particular instance of the methodology also represents a generalization of nonlinear spatial interpolation of daily streamflow using FDCs. Rather than relying on single index stations, as is commonly done to reflect streamflow timing, this approach to simulation leverages geostatistical tools to allow a region of neighbors to reflect streamflow timing.

## 1  Introduction

Simulation of historical daily streamflow at ungauged locations is one of the grand challenges of the hydrological sciences (Sivapalan, 2003; Sivapalan et al., 2003; Hrachowitz et al., 2013; Parajka et al., 2013). Over the past 20 years, at least, research into simulation of historical streamflow has increased. In addition to ongoing international efforts, the U.S. Geological Survey has embarked upon a National Water Census of the United States (Alley et al., 2013), which seeks to quantify hydrology

across the country to provide information to help improve water use and security. However, regardless of the method used for the simulation, uncertainty will always remain and may result in some distributional bias (Farmer and Vogel, 2016). The objective of this work is to present a technique to correct for bias in the magnitudes of a streamflow simulation. While the mechanics of this technique are not novel, the novelty of this work lies in the generalization of this technique for use in bias

correction. The method is intended for use at ungauged sites and an idealized experiment is constructed to demonstrate both the potential utility and one example of realized utility.

As defined here, distributional bias in simulated streamflow is an error in reproducing the tails of streamflow distribution. As attested to by many researchers focused on the reproduction of historical streamflow, this bias commonly appears as a general overestimation of low streamflow and underestimation of high streamflow (Skøien and Blöschl, 2007; Rasmussen et al., 2008;

Farmer et al., 2014, 2015; Farmer, 2016; Farmer and Vogel, 2016; Archfield et al., 2010, 2013). The result is an effective squeezing of the streamflow distribution, bringing the tails of the distribution closer to the central values. This distributional squeezing is often most notable in the downward bias of extreme high-flow events (as in, e.g., Lichty and Liscum, 1978; Thomas, 1982; Sherwood, 1994). Bias of high streamflows is particularly concerning, as examinations of extreme high-flow events are a common and influential use of historical simulation and long-term (decadal) forecast. Consider, for example, the

motivation for work by Archfield et al. (2013): As simulated streamflows were being routed through a reservoir operations model for flood mitigation, large bias in high streamflows would have severely affected resulting decisions. Of course, this tendency towards distributional compaction is not a universal truth that occurs without variation; the resulting bias will vary widely depending on the structure of the residuals (Farmer and Vogel, 2016).

Because of the importance of accurately representing extreme events, it is necessary to consider how the distributional bias

of streamflow simulations can be reduced. The approach presented here assumes that, while the streamflow magnitudes of a historical simulation are biased, the temporal structure or rank-order of simulated streamflows is relatively accurate. The nature of this approach is predicated on an assumption that although a historical simulation may produce a distribution of streamflow with biased tails, the temporal sequence of relative rankings or nonexceedance probabilities of the simulated streamflow retains valuable information. With this assumption, it can be hypothesized that distributional bias can be reduced, while not negatively

impacting the overall performance, by applying a sufficiently accurate independently estimated representation of the period-of-record flow duration curve (FDC) to rescale each streamflow value based on the streamflow value of the regional FDC for the corresponding nonexceedance probabilities (see Material and Methods, below).

The approach presented here can be perceived as a generalization of the nonlinear spatial interpolation of daily streamflow using FDCs as conceived by Fennessey (1994) and Hughes and Smakhtin (1996) and widely used thereafter (Smakhtin, 1999;

Mohamoud, 2008; Archfield et al., 2010; Shu and Ouarda, 2012). As traditionally applied, nonlinear spatial interpolation proceeds by simulating nonexceedance probabilities at a target location using a single neighboring streamgage (though Hughes and Smakhtin (1996) recommend and Shu and Ouarda (2012) test the use of multiple streamgages) and then interpolating those nonexceedance probabilities along a FDC. The approach tested here seeks to bias-correct a simulated time series of daily discharge using an independently estimated FDC, and, when viewed in another way, presents a novel form of nonlinear spatial

interpolation.

Furthermore, though necessarily explored in this study through the use of a single technique for hydrograph simulation, this approach may be a means to effectively bias-correct any simulation of streamflow, including those from rainfall-runoff models, as presented by Pugliese et al. (2017). Pugliese et al. (2017) used a geostatistical tool to produce site-specific FDCs and then used this information to post-process simulated hydrographs from a deterministic model. Though the underlying methods of producing the FDC and simulated hydrograph are different, the approach proposed by Pugliese et al. (2017) is the same as that explored here. Further discussion of the relationship of the approach presented here to others in the field is provided below.

The remainder of this work is organized in the following manner. The section titled "Material and Methods" provides a description of the retrieval of observed streamflow, the estimation of simulated streamflows, the calculations of observed FDCs, the estimation of simulated FDCs, and the application and evaluation of the bias-correction. A section titled "Results" follows and it documents the bias in the original simulated streamflows, analyzes the potential bias correction that could be achieved if it were possible to know the observed FDC at an ungauged location, and the bias correction that would be realized through an application of regional regression. The following section titled "Discussion" considers the implications of these results and hypothesizes how the methodology might be applied and improved. The major findings of this work are then summarized in the section titled "Summary and Conclusions".

## 2 Material and Methods

This section, which is divided into four subsections, provides a description of the methods applied here. The first subsection describes the collection of observed streamflow as well as the initial simulation of streamflow. As the approach used here is applicable to any simulated hydrograph, the details of hydrograph simulation are not exhaustively documented. Instead, beyond a brief introduction, the reader is directed to relevant citations, as no modifications to previous methods are introduced here. The second subsection discusses the use of regional regression to define independently estimated FDCs. Again, as any method for the estimation of FDCs could be used and this application is identical to previously reported applications, following a brief introduction, the reader is directed to the relevant citations. The third subsection provides a description of how bias correction was executed, and the fourth subsection describes how the performance of this approach to bias correction was assessed.

### 2.1 Observed and Simulated Streamflow

The proposed approach was explored using daily mean streamflow data from the reference-quality streamgages included in the GAGES-II database (Falcone, 2011) within the conterminous United States for the period from October 01, 1980, through September 30, 2013. To allow for the interpolation, rather than extrapolation, of all quantiles considered later, streamgages were screened to ensure that at least fourteen (14) complete water years (October 01 through September 30) were available for each record considered; 1168 such streamgages were available. The selected reference streamgages are indicated in Figure 1. The streamflow data were obtained directly from the website of National Water Information System (NWISWeb, http://waterdata.usgs.gov, accessed 20 Sept. 2017). For each streamgage, associated basin characteristics were obtained from the GAGES-II database (Falcone, 2011).

To control for streamflow distributions that vary over orders of magnitude, the simulation and analysis of streamflow at these streamgages is best explored through the applications of logarithms. To avoid the complication of taking the logarithm of a zero, a small value was added to each streamflow observation. The U.S. Geological Survey rounds all mean daily streamflow to two decimal places in units of cubic feet per second (cfs, which can be converted to cubic meters per second using a factor of 0.0283). As a result, any value below 0.005 cfs is rounded to and reported as 0.00 cfs. Because of this rounding procedure, the small additive value applied here was 0.0049 cfs. While there may be some confounding effect produced by the use of an additive adjustment, as long as this value is not subtracted on back transformation, the following assessment of bias and bias correction will remain robust. That is, rather than evaluating bias in streamflow, technically this analysis is evaluating the bias in streamflow plus a correction factor. The conclusions remain valid as the assessment still evaluates the ability of a particular method to remove the bias in the simulation of a particular quantity.

Though the potential for distributional bias applies to any hydrologic simulation (Farmer and Vogel, 2016), for this study, initial predictions of daily streamflow values for each streamgage were obtained by applying the pooled ordinary kriging approach (Farmer, 2016) to each 2-digit Hydrologic Unit (figure 1) through a leave-one-out cross-validation procedure on the streamgages within the 2-digit Hydrologic Unit. The Hydrologic Unit system is a common method for delineating watersheds in the U.S. As described by Seaber et al. (1987), the 2-digit Hydrologic Units, or regions (as seen in Figure 1), roughly align with the major river basins of the U.S. This approach considers all pairs of common-logarithmically transformed unit streamflow (discharge per unit area) at each day and builds a single, time-invariant semivariogram model of cross-correlation that is then used to estimate ungauged streamflow as a weighted summation of all contemporary observations. A spherical semivariogram was used as the underlying model form. Additional information on the time series simulation procedure is provided by Farmer (2016). Note that the choice of pooled ordinary kriging is only made as an example of a streamflow simulation method; it is not implied that the bias observed or methods applied are relevant only to this approach to simulation. Because the novelty of this work is in the application of bias correction, further details on the particular simulation method employed are left for the reader to investigate in the cited works (Farmer, 2016).

## 2.2 Estimation of Flow Duration Curves

Daily period-of-record FDCs were developed independently of the streamflow simulation procedure by following a regionalization procedure similar to that of Farmer et al. (2014) and Over et al. (2018). Observed FDCs were obtained by determining the percentiles of the streamflow distribution across complete water years between 1981 and 2013 using the Weibull plotting position (Weibull, 1939). Twenty-seven percentiles with exceedance probabilities of 0.02%, 0.05%, 0.1%, 0.2%, 0.5%, 1%, 2%, 5%, 10%, 20%, 25%, 30%, 40%, 50%, 60%, 70%, 75%, 80%, 90%, 95%, 98%, 99%, 99.5%, 99.8%, 99.9%, 99.95%, and 99.98% were considered. The selection of streamgages with at least 14 complete water years ensures that all percentiles can be calculated from the observed data. These percentiles derived from the observed hydrograph represent the "unknowable observation" in an application for prediction in ungauged basins. Therefore, to simulate the truly ungauged case, these same percentiles were estimated using a leave-one-out cross-validation of regional regression.

A regional regression across the streamgages in each 2-digit Hydrologic Unit of each of the 27 FDC percentiles was developed using best-subsets regression. Best-subsets regression is common tool for exhaustive exploration of the space of potential explanatory variables. All models with a given number of explanatory variables are computed, exploring all combinations of variables. The top models for a given number of explanatory variables are then identified by a performance metric like the Akaike Information Criterion. This is repeated for several model sizes to fully explore the possibilities for variables and regression size. For each regression, the drainage area was required as an explanatory variable. At a minimum, one additional explanatory variable was used. The maximum number of explanatory variables was limited to the smaller of either six explanatory variables or 5% of the number of streamgages in the region, rounded up to the next larger whole number. The maximum of six arises from what is computationally feasible for the best subsets regression function used, whereas the maximum of 5% of streamgages was determined from a limited exploration of the optimal number of explanatory variables as a function of the number of streamgages in a region. Explanatory variables were drawn from the GAGES-II database (Falcone, 2011). As documented by Farmer et al. (2014) and Over et al. (2018), a subset of the full GAGES-II dataset was chosen to avoid strong correlations. As the focus of this work is not on the estimation of the FDCs, the reader is referred to the works of (Farmer et al., 2014, 2018) and Over et al. (2018) to explore the exact procedures.

In order to allow different explanatory variables to be used to explain percentiles at different streamflow regimes, the percentiles were grouped into a maximum of three contiguous streamflow regimes based on the behavior of the unit FDCs (i.e., the FDCs divided by drainage area) in the 2-digit Hydrologic Units. The regimes are contiguous in that only consecutive percentiles from the list above can be included in the same regime; the result is a maximum of three regimes that can be considered "high", "medium" and "low" streamflows, though the number of regimes may vary across 2-digit Hydrologic Units. The percentiles in each regime were estimated by the same explanatory variables, allowing only the fitted coefficients to change. The final regression form for each regime was selected by optimizing the average adjusted coefficient of determination, based on censored Gaussian (Tobit) (Tobin, 1958) regression to allow for values censored below 0.005 cfs, across all percentiles in the regime. The addition of a small value was used to avoid the presence of zeros and enable a logarithmic transformation, but this does not avoid the problem of censoring. Censoring below the small value added must still be accounted for so that smaller numbers do not unduly affect the regression. This approach to percentile grouping was found to provide reasonable estimates while minimizing the risk of non-monotonic or otherwise concerning behavior. Further details on this methodology can be found in the associated data and model archive (Farmer et al., 2018) and in Over et al. (2018).

When estimating a complete FDC as realized through a set of discrete points, non-monotonic behavior is likely (Poncelet et al., 2017). If the regression for each percentile were estimated independently, non-monotonicity would be almost unavoidable. By using three regimes and keeping the explanatory variables the same within each, the potential for non-monotonicity is reduced. The greatest risk of non-monotonic behavior occurs at the regime boundaries. If the FDC used to bias-correct is not perfectly monotonic, the effect will be to alter the relative timing of streamflows. While it would be ideal to avoid any risk of non-monotonic behavior, it is a rather difficult task. An alternative might be to consider the FDC as a parametric function, but Blum et al. (2017) demonstrate how difficult this can be for daily streamflows. Of course, the use of regional regression is not the only tool for estimating an FDC (for reviews, see Castellatin et al., 2013; Pugliese et al., 2014, 2016).

## 2.3 Bias Correction

To implement bias correction, the initial predictions of the daily streamflow values by the ordinary kriging approach were converted to streamflow nonexceedance probabilities using the Weibull plotting position (Weibull, 1939). The nonexceedance probabilities were then converted to standard normal quantiles and linearly interpolated along an independently estimated FDC. For the linear interpolation, the independently estimated FDC was represented as the standard normal quantiles of the associated nonexceedance probabilities versus the common logarithmic transformation of the streamflow percentiles. In the case where the standard normal quantile being estimated from the simulated hydrograph was beyond the extremes of the FDC, the two nearest percentiles were used for linear extrapolation. In this way, the ordinary kriging simulations were bias-corrected, based on the assumption that the simulated volumes are less accurate than the relative ranks of the simulated values, by rescaling the simulated volumes to an independently estimated FDC. By changing the magnitudes of the simulated streamflow distribution, this approach rescales the distribution of the simulated streamflow.

Figure 2 provides a simplified representation of this bias-correction methodology. Starting in the upper-left panel and proceeding clockwise, after simulating the hydrograph with a given methodology (pooled ordinary kriging was used here), the resulting streamflow value on a given day can be converted to appropriate non-exceedance probabilities by proceeding from point A, through point B and down to point C. Moving then from point C to point D maps the estimated non-exceedance probability onto an independently estimated FDC. Finally, the streamflow value produced at point D is mapped to the original date (point E) to reconstruct a bias-corrected hydrograph. Note that this is a simplified description: as described above a slightly more complex interpolation procedure is used for the FDCs represented by a set of discrete points.

As can be seen in Figure 2, this methodology is quite similar to that conceived by Fennessey (1994) and Hughes and Smakhtin (1996). The novelty of this work lies in its application. That is, both Fennessey (1994) and Hughes and Smakhtin (1996) imagine a case where the original hydrograph from which non-exceedance values will be drawn (upper-left panel of Figure 2) is drawn from an index station of some sort; here the temporal structure could be drawn from any technique for at-site hydrograph simulation. This generalization allows bias-correction of any hydrograph simulation.

## 2.4 Evaluation

The hypothesis of this work, that distributional bias in the simulated streamflow can be corrected by applying independently estimated FDCs, was evaluated by considering the performance of these bias-corrected simulations at both tails of the distribution. The differences in the common logarithms of both high and low streamflow were used to understand and quantify the bias (simulation minus observed) and the correction thereof. That is,

$$bias_s = \frac{\sum_{i=1}^{n} (\log_{10}(\hat{Q}_{s,i}) - \log_{10}(Q_{s,i}))}{n} \tag{1}$$

where $s$ indicates the site of interest, $\hat{Q}$ indicates the predicted streamflow, whether the original simulation or the bias-corrected simulation, $Q$ indicates the observed streamflow, and $n$ indicates the number of values being assessed. This difference can be approximated as a percent by computing ten to the power of the difference and subtracting one from this quantity (Eng et al.,

2009):

$$bias_{s,\%} = 100 \cdot (10^{bias_s} - 1) \tag{2}$$

The differences in the root mean squared error of the common logarithms of the predicted streamflow were used to quantify improvements in accuracy. The root mean squared error of the common logarithms of streamflow is calculated as

$$5 \quad rmsel_s = \sqrt{\frac{\sum\limits_{i=1}^{n}(\log_{10}(\hat{Q}_{s,i}) - \log_{10}(Q_{s,i}))^2}{n}} \tag{3}$$

Improvements in accuracy may or may not occur when bias is reduced. The significance of both these quantities, and the effects of bias correction on these quantities, was assessed using a Wilcoxon signed rank test (Wilcoxon, 1945). For assessments of bias, the null hypothesis was that the bias was equivalent to zero. For assessments of the difference in bias or accuracy with respect to the baseline result, the null hypothesis was that this difference was zero.

Distributional bias and improvement of that bias were considered in both the high and low tails of the streamflow distribution. Two methods were used to capture the bias in each tail. One method, referred to herein as an assessment of the observation-dependent tails, considers the observed nonexceedance probabilities to identify the days on which the highest and lowest 5% of streamflow occurred. For each respective tail, the errors were assessed based on the observations and simulations of those fixed days. The other method, referred to herein as an assessment of the observation-independent tails, compares the ranked

top and bottom 5% of observations with the independently ranked top and bottom 5% of simulated streamflow. Errors in the observation-dependent tails are an amalgamation of errors in the sequence of nonexceedance probabilities (the temporal structure) and in the magnitude of streamflow, whereas errors in the observation-independent tails reflect only bias in the ranked magnitudes of streamflow. In the same fashion, evaluation of the complete hydrograph can be assessed sequentially (sequential evaluation), retaining the contemporary sequencing of observations and simulations, or distributionally (distributional evalua-

tion), considering the observations and simulations ranked independently. Though the overall accuracy will vary between the sequential and distributional case, overall bias will be identical in both cases.

     With an analysis of both observation-dependent and observation-independent tails, it is possible to begin to tease out the effect of temporal structure on distributional bias. The bias in observation-independent tails is not directly tied to the temporal structure, or relative ranking, of simulated streamflow. That is, if the independently estimated FDC is accurate, then even

if relative sequencing of streamflow is badly flawed, the bias-correction of observation-independent tails will be successful. However, even if the distribution is accurately reproduced after bias correction, the day-to-day performance may still be poor. For observation-dependent tails, the temporal structure plays a vital role on the effect of bias correction. If the temporal structure is inaccurate in the underlying hydrologic simulation, then the bias correction of observation-dependent tails will be less successful.

The bias correction approach was first tested with the observed FDCs. These observed FDCs would be unknowable in the truly ungauged case, but this test allows for an assessment of the potential utility of this approach. This examination is followed by an application with the regionally regressed FDCs described above, demonstrating one realization of this generalizeable

method. This general approach to bias correction could be used with other methods for estimating the FDC and could also be used with an observed FDC for record extension, though neither of these possibilities are explored here.

## 3 Results

Figures 3 and 4 show the overall bias and accuracy of the reproduced hydrographs; these figures are quantified in Tables 1 and 2. Figure 5 and Table 1 summarize the tail bias in all approaches to streamflow simulation considered here. Similarly, Figure 6 and Table 2 summarize the tail accuracy of all approaches. These results are discussed in detail below, beginning with a discussion of the bias and accuracy in the original kriged simulations. This is followed by a consideration of the effectiveness of bias-correction with observed FDCs as emblematic of the theoretical potential of this approach. The realization of this theoretical potential through the regionally regressed FDCs is subsequently presented. Complete results can be explored and reproduced using the associated model and data archive (Farmer et al., 2018).

### 3.1 Simulated Hydrographs without Correction

There is statistically significant overall bias at the median (-7.1%; $10^{-0.0318} - 1$) in the streamflow distribution simulated by the kriging approach applied here (Figure 3, boxplot A), but more significant bias is apparent in the upper and lower tails of the distribution (Figure 5, boxplots A, D, G and J). Both the observation-dependent and observation-independent upper tails of the streamflow distribution demonstrate significant downward bias (Figure 5, boxplots D and J). At the median, the observation-dependent upper tail is underestimated by approximately 38% (Table 1, row 1; Figure 5, boxplot D), while the observation-independent upper tail is underestimated by approximately 23% (Table 1, row 2; Figure 5, boxplot J). For the lower tail, the observation-dependent tail shows a median overestimation of 36% (Table 1, row 1; Figure 5, boxplot A), while the observation-independent tail is underestimated by less than one percent (table 1, row 2; Figure 5, boxplot G). The bias is much more variable, producing greater magnitudes of bias more often, in the lower tails than in the upper tails. Generally, biases in the observation-independent tails are less severe, both in the median and in range, than those in the observation-dependent tails. To provide some information on regional performance and incidence of bias, Figure 7 shows the spatial distribution of bias in each tail (discussion of this distribution is provided below).

In both observation-dependent and –independent cases, downward bias in the upper tail is more probable than upward biases in the lower tail. For the observation-dependent tails, approximately 89% of streamgages show downward bias for the upper tail (Figure 5, boxplot D), and approximately 61% of the streamgages upward bias in the lower tail (Figure 5, boxplot A). For the observation-independent tails, approximately 80% of streamgages show downward bias in the upper tail (Figure 5, boxplot J) and approximately 50% of the streamgages exhibit upward bias in the lower tail (Figure 5, boxplot G), indicating, as does the small median bias value, that the lower tail biases are relatively well balanced around zero for the observation-independent case for these simulations.

With respect to their central tendencies, these results show upward bias in lower tails and downward bias in upper tails of the distribution of streamflows from the original simulations for both observation-dependent and observation-independent cases.

There is, of course, a great degree of variability around this central tendency. With these baseline results, the bias-correction method presented here seeks to mitigate these biases.

## 3.2 Bias Correction with Observed FDCs

The results for this idealized case that could not be applied in practice provide clear evidence that distributional bias in simulated streamflow can be reduced by rescaling using independently estimated FDCs. This evidence is apparent in the reduction of the magnitude and variability of overall bias (Figure 3, boxplot C; Table 1, rows 5 and 6) and of the bias in the observation-independent tails of the streamflow distribution (Figure 5, boxplots I and L) when observed FDCs are used for rescaling. Similarly, the overall distributional accuracy is much improved (Figure 4, boxplot F; Table 2, rows 5 and 6), as is the accuracy of observation-independent tails (Figure 6, boxplot I and L). The effect on observation-dependent tails (Figure 5, boxplots C and F) and overall sequential accuracy (Figure 4, boxplot C) is less compelling but still substantial.

Whereas the measures of bias and accuracy are summarized in Tables 1 and 2, Tables 3 and 4 summarize the change in absolute bias and in accuracy, respectively. With the use of observed FDCs, the overall bias is reduced to a tenth of a percent at the median (Table 1, rows 5 and 6). This represents a significant median reduction of 0.14 common-logarithm units in the overall absolute bias (Table 3, rows 3 and 4). Overall, the distributional accuracy is improved by a median of 0.21 common-logarithm units (Table 4, row 4). Of all streamgages considered, 99% saw a reduction in the overall absolute bias, and all saw improvements in overall distributional accuracy. These improvements extend to both observation-independent tails of the distributions. The lower observation-independent tails have a median 0.35 common-logarithm unit reduction in absolute bias (Table 3, row 4). For the upper tail, the median reduction in absolute bias is 0.14 common-logarithm units (Table 3, row 4). Nearly all streamgages (99%) saw reduction in absolute bias of the observation-independent tails. Table 4 (row 4) shows similar improvements in tail accuracy: -0.37 and -0.15 units in the lower and upper tails, respectively, with nearly all streamgages (excepting the lower tail of a single streamgage, likely the result of the interpolation procedure) showing improved tail accuracy.

With the use of a perfect, observed FDC for bias correction, one would expect that nearly all bias would disappear, but the results do not show this. The temporal structure of the simulated hydrograph continues to play a role in the bias of observation-dependent tails. The observation-independent tails continues to exhibit a small degree of residual bias, though it is still slightly non-intuitive. This residual bias arises from the effect of representing the FDC as a set of discrete points and interpolating between them. There may be some additional effect from the small value added to avoid zero-valued streamflows or the censoring procedure, but initial exploration found little impact.

The overall sequential performance (Figure 4, boxplot C) and the performance of observation-dependent tails (Figures 5 and 6, boxplots C and F) demonstrate the degree to which errors in the temporal structure result in bias in the observation-dependent case even when observed FDCs are used for bias correction. Both the observation-dependent lower and upper tails exhibit bias: 30% and -20%, respectively, at the median (Table 1, row 5, with transformation using Eq. 2). Absolute bias in both tails show median reductions; sequential accuracy and observation-dependent tail accuracy are also improved at the median (Tables 3 and 4, row 3). Proportionally, 82% of the observation-dependent lower tails and 86% of the observation-dependent

upper tails showed reduction in absolute bias (Figure 5, boxplots C and F); 85% of observation-dependent lower tails and 79% of observation-dependent upper tails showed improvements in accuracy (Figure 6, boxplots C and F). Despite improvements in overall bias and accuracy from rescaling with observed FDCs, the residual bias in the observation-dependent lower tail (Figure 5, boxplot C) is almost always positive (upward bias) and upper tails (Figure 5, boxplot F) are almost negative (downward bias), a result which arises primarily from errors in the simulated temporal structure.

To understand the effect of errors in the temporal structure further, consider Figure 8, which shows the mean error in the nonexceedance probabilities, i.e., the difference in the ranks of the observed and simulated streamflows, of the observation-dependent upper and lower tails. The nonexceedance percentages in the lower tail are overestimated by a median of 3.8 points with 5th and 95th percentiles of 0.9 and 20.5, while the percentages in the upper tail are underestimated by 2.4 points, with 5th and 95th percentiles of -0.5 and -12.6 points. The distributions of errors in the nonexceedance probabilities closely reflect the distribution of bias in the observation-dependent tails (Figure 5, boxplots C and F). These results show that the inaccuracy in the nonexceedance probabilities (i.e., errors in temporal structure) will obscure, at least partially, the improvement offered by bias correction when considering the observation-dependent errors, even when an observed FDC is used for bias correction. These errors in temporal structure also almost always result in errors in a particular direction - low for high flow and high for low flows.

## 3.3 Bias Correction with Regionally Regressed FDCs

When the uncertainty of regionally regressed FDCs is introduced into the bias correction procedure, the potential value of the bias correction procedure is not as convincing. There is a slight, but significant, increase in the overall bias (Table 3, rows 1 and 2). Whereas the original estimated streamflow displays a median bias of approximately 7.1%, the median overall bias is approximately 7.6% after bias correction with estimated FDCs, (Table 1, rows 3 and 4). Though statistically significant, the distribution of bias does not appear to have changed in a meaningful way (Figure 3, boxplots A and B). The overall accuracy, sequential and distributional, is also degraded (Figure 4, boxplots B and E; Table 4, rows 3 and 4), with more than 60% of streamgages showing degradation in sequential and distributional accuracy.

The observation-independent tails, which are not affected by errors in temporal structure, show a divergence in performance between the results obtained using observed FDCs and those obtained using regionally regressed FDCs. With observed FDCs, both tails demonstrated substantial reductions in absolute bias and improvements in accuracy. With regionally regressed FDCs, the upper observation-independent tails continue to show reductions in absolute bias (Table 3, row 2; Figure 5, boxplots J and K) and improvements in accuracy (Table 4, row 2; Figure 6, boxplots J and K), while the lower observation-independent tails show a significant increase in absolute bias (Table 3, row 2; Figure 5, boxplots G and H) and a degradation of accuracy (Table 4, row 2; Figure 6, boxplots G and H). After bias correction with regionally regressed FDCs, only 44% of observation-dependent lower tails showed reductions in absolute bias; 58% of upper tails showed reductions in absolute bias.

The effects of the rescaling with FDCs estimated with regional regression on overall and observation-independent tail bias and accuracy can be better understood if the properties of the estimated FDCs are considered. Figure 9 shows the bias (left panel) and accuracy (right panel) of the lower and upper tails of the regionally regressed FDCs. Recall that the estimated

FDCs are composed of 27 quantiles, of which the upper and lower tails contain only the eight values with nonexceedance probabilities 95% and larger and 5% and smaller, respectively. The upper tails are reproduced through regional regression with an insignificant 2.5% median downward bias, but the lower tails exhibit a significant negative median bias of 38.35% (Table 1, row 7). Because of this bias in the lower tail of the regionally regressed FDCs, the regionally regressed FDCs are unable to correct the bias in the simulated hydrograph, instead turning a small median bias into large one. As there is no temporal uncertainty in the observation-independent tails, the resulting bias arises from the bias of the regionally regressed FDC. Illustrating this fact, the -38% bias in the lower tail of the regionally regressed FDC approximates the -33% in the observation-independent lower tail, while the -2.5% bias in the upper tail of the regionally regressed FDC approximates the -3.7% bias in the observation-independent upper tail. The introduction of this additional bias, beyond failing to correct any underlying bias in the simulated hydrograph, also markedly increased the variability of both bias and accuracy.

The results are similar for the observation-dependent tails produced after bias correction with regionally regressed FDCs, even when complicated by the addition of temporal uncertainty as discussed in section 3.2 with reference to Figure 8. In some cases, the errors in the temporal structure (nonexceedance probability) counteract the additional bias from regionally regressed FDCs. For example, the observation-dependent lower tails have a median bias of 13%, which possesses a smaller magnitude and different sign than the median -33% bias seen in the observation-independent lower tail (Table 1, rows 3 and 4). The addition of temporal uncertainty actually reduced the increase in absolute bias (Table 3, rows 1 and 2) and reduced the degradation of accuracy in the lower tail (Table 4, rows 1 and 2). These slight improvements result from an offsetting of the underestimated regionally regressed FDCs by the overestimated nonexceedance probabilities. While interesting, it seems unlikely that this result can be generalized in a simple way: that is, the errors in estimated FDCs cannot be expected to balance out the errors in nonexceedance probabilities without deleterious effects on other properties. To this point, as noted, rescaling by these regionally regressed FDCs with underestimated lower tails result in similarly underestimated observation-independent lower tails.

The introduction of uncertainty from regionally regressed FDCs diminishes the advantages gained by biased correction with observed FDCs. Considering the observation-independent lower tails, 55% of streamgages show reductions in absolute bias with observed FDCs that were reversed into increases of absolute bias by the introduction of regionally regressed FDCs. Another 43% of streamgages show smaller reductions in absolute bias when observed FDCs were replaced with regionally regressed FDCs. For the observation-dependent lower tails, 37% of streamgages have reversals and 31% show smaller reductions in absolute bias. For the observation-independent upper tails, 41% show reversals and 56% yield smaller reductions in absolute bias. For the observation-dependent upper tails, 24% produce reversals and 40% provide smaller reductions in absolute bias. Results are similar with respect to accuracy: while many streamgages saw reversals, a large proportion of streamgages continue to demonstrate improvements.

## 4 Discussion

Though the first analysis presented, which utilized observed FDCs for bias correction, represents only an assessment of hypothetical potential of this general approach, the approach to bias correction presented here produced near universal and substantial reduction in bias and improvements in accuracy, overall and in each tail, for both observation-dependent and –independent evaluation cases when the uncertainty in independently estimated FDCs was minimized. For the observation-independent evaluation case, the errors are removed almost completely, and the remaining errors in the observation-dependent case mimic the temporal structure (nonexceedance probability) errors. These results, which are not applicable under the conditions of the true ungauged problem, demonstrate that the bias-correction approach introduced here is theoretically valid. However, this improvement becomes inconsistent with respect to bias and generally reduces the accuracy when the biased and uncertain regionally regressed FDCs are used. Furthermore, in both the observation-dependent and observation-independent tails in the case of rescaling by regionally regressed FDCs, the improvements in the lower tails are much more variable than the improvements in the upper tail (Figures 5 and 6; Tables 3 and 4). This result is not surprising, given the more-variable nature of lower-tail bias and accuracy (Figures 5 and 6).

The regional regressions developed here were much better at estimating the upper tail of the streamflow distribution than estimating the lower tail. This provides a convenient comparison: the bias correction of lower tails with regionally regressed FDCs only improved the bias in the observation-dependent case when the low bias of the regionally regressed FDC offset the high bias of the observation-dependent tails, and did not improve accuracy in either case. However, the bias correction of upper tails with regionally regressed FDCs, which produced the upper tails with much less bias, continued to show, like in the case of observed FDCs, improvements in bias and accuracy, though to a much smaller degree than the improvements produced by observed FDCs.

Particularly in the lower tail of the distribution, the effectiveness of this bias-correction method is strongly influenced by the accuracy of the independently estimated FDC. The change in the absolute bias of the observation-independent lower tail has a 0.72 Pearson correlation with the absolute bias of the lowest eight percentiles of the FDC estimated with regional regression, showing that the residual bias in the FDC of the bias-corrected streamflow simulations is strongly correlated with the bias in the independently estimated FDC. The analogous correlation for the upper tail is 0.31. For the observation-dependent these correlations are only 0.33 for each tail, the reduced correlation for the lower tail being a result of the combination of the uncertainty in the temporal structure and in the regionally regressed FDC. Therefore, as regional regression is not the only tool for estimating FDCs (for other examples, see Castellatin et al., 2013; Pugliese et al., 2014, 2016), improved methods for FDC estimation would further increase the impact of this bias-correction procedure. There are also hints that the representation of the FDC as a set of discrete points degraded performance. Further work might address the question of improving FDC simulation. Still further, seasonal FDCs or some other methods of increasing the temporal variability of FDCs could improve performance of this general bias correction approach.

While this method of bias correction, as implemented here using regionally regressed FDCs, improves the bias in the upper tails, it had a negative impact on lower tails. This makes the question of application or recommendation more poignant. Under

what conditions might this approach be worthwhile? Initial exploration did not find a strong regional component to performance of the bias correction method. Figure 7 shows the original tail bias from pooled ordinary kriging; at each point the accuracy of the bias correction method is dependent on the original bias present as well as the error in the independently estimated FDC. For some regions, like New England, where FDCs are well estimated by regional regression, there is a general improvement in accuracy under bias correction with regionally regressed FDCs, but the improvement is highly variable. Instead, the strongest link is with the reproduction of the FDC. When magnitude of tail biases of the regionally regressed FDC was under 20%, more than 50% of streamgages showed improvements in bias, both overall and in the tails of the distribution. At a particular ungauged site, it may not always be possible to determine the accuracy with which a given FDC estimation technique might perform beyond a regional cross-validated assessment of general uncertainty, making it difficult to determine if these results can be generalized. If accuracy of the estimated FDCs can be estimated, it may also be useful to consider rescaling one tail and not the other, depending on the estimated accuracy. Further work might explore the effects of hydroclimates on the ability to reproduce reliable FDCs with which to implement this bias-correction procedure.

The results of this work were also discussed in reference to earlier work that suggested a prevalence, though not a universality, of underestimation of high streamflows and over-estimation of low streamflows. Similarly, the bias correction approach produced a wide variability of results; where the high tails might have been improved, the lower tails might have been degraded. Figure 10 shows the correspondence of tails across all sites. While there is a move towards unbiasedness at some sites (along the vertical axis), there is a great degree of variability that makes it difficult to draw general conclusions. In some situations, as in Panel D, the variability may actually be increasing with bias correction. Though all methods will produce variability, it remains to future research to determine if a more consistent representation of the FDC might reduce the variability of this performance.

When looked at from the point-of-view of the estimated FDCs that need temporal information in order to simulate stream-flow, this approach to bias correction is as akin to an extension of the non-linear spatial interpolation using FDCs developed by Fennessey (1994) and Hughes and Smakhtin (1996) as it is bias correction. Here it is approached as a method for bias correction, but it can also be thought of as a novel approach to simulate the nonexceedance probabilities at an ungauged location to be used with estimated distributional information (FDCs) to simulate streamflow. In the early uses of nonlinear spatial interpolation using FDCs, the simulated nonexceedance probabilities were obtained from a hydrologically appropriate neighboring or group of neighboring streamgages (Shu and Ouarda, 2012), though the approach to identifying a hydrologically appropriate neighbor has varied. Here, the entire network is used to approximate the ungauged nonexceedance probabilities, much like the indexing problem was overcome with ordinary kriging of streamflow directly (Farmer, 2016). Two major sources of uncertainty are inherent in nonlinear spatial interpolation using FDCs: uncertainty in the nonexceedance probabilities and uncertainty in the FDC. This work addresses the general approach by attacking the former and observing that performance may be further limited by the latter. The potential success of this approach to bias correction is likely not specific to simulation with ordinary kriging.

That this approach to bias correction does improve the observation-dependent tails and the overall performance when observed FDCs are used shows that the temporal structure of the underlying simulation retains useful information even if the tails

of the original simulation are biased. However, some error remains in the simulated nonexceedance probabilities. A natural extension would be to investigate if it might be more reasonable to estimate nonexceedance probabilities directly rather than extracting their implicit values from the estimated streamflow time series as was done here. Here, the nonexceedance probabilities were derived from a simulation of the complete hydrograph. In this alternative approach, the discharge volumes would not be estimated but rather only the daily nonexceedance probabilities. Farmer and Koltun (2017) executed a kriging approach to estimate daily nonexceedance probabilities in a smaller data set in Ohio. They found that modeling probabilities directly resulted in similar tail biases of nonexceedance probability to that observed when, as in Farmer (2016), simulating streamflow directly. In earlier work, Farmer (2015) showed that kriging nonexceedance probabilities directly and then redistributing them via an estimated FDC, as compared with kriging streamflow directly, had only a marginal effect on bias in the tails. Further exploration of this question, whether to estimate nonexceedance probabilities directly or derive them from streamflow simulations, is left for future research. This current study focuses on the more general question of whether the distributional bias in a set of simulated streamflow, the provenance thereof being more or less inconsequential, could be reduced using a regionally regressed FDC.

As mentioned earlier, recent work by Pugliese et al. (2017) explore how this generalization of non-linear spatial interpolation using FDCs can be used to improve simulated hydrographs produced by a continental scale deterministic model. They consider it as an approach to inform a large-scale model with local information, thereby improving local application without further calibration. In 46 basins in Tyrol, Pugliese et al. (2017) saw universal improvement in the simulated hydrographs, though they did not explore tails biases. The results presented here provide an analysis across a wider range of basin characteristics and climates, demonstrating a link between how well the FDC can be reproduced and ultimate improvements in performance or reductions in bias.

Although the results presented here are promising, they demonstrate that the performance of two-stage modeling, where temporal structure and magnitude are largely decoupled, is limited by the less well performing stage of modeling. In this case, alternative methods for estimating the FDC might prove worthwhile (e.g., see Castellatin et al., 2013; Pugliese et al., 2014, 2016).

## 5   Summary and Conclusions

Regardless of the underlying methodology, simulations of historical streamflow often exhibit distributional bias in the tails of the distribution of streamflow, usually an overestimate of the lower tail values and an underestimate of the upper tail values. Such bias can be extremely problematic, as it is often these very tails that affect human populations and other water management objectives the most and, thus, these tails that receive the most attention from water resources planners and managers. Therefore, a bias-correction procedure was conceived to rescale simulated time series of daily streamflow to improve simulations of the highest and lowest streamflow values. Being akin to a novel implementation of nonlinear spatial interpolation using flow-duration curves, this approach could be extended to other methods of streamflow simulation.

In a leave-one-out fashion, daily streamflow were simulated in each 2-digit hydrologic unit code using the pooled ordinary kriging. Regional regressions of 27 percentiles of the flow-duration curve in each 2-digit hydrologic unit code were independently developed. Using the Weibull plotting position, the simulated streamflow were converted into nonexceedance probabilities. The nonexceedance probabilities of the simulated streamflow were used to interpolate newly simulated streamflow volumes from the regionally regressed flow-duration curves. Assuming that the sequence of relative magnitudes of streamflow retains useful information despite possible biases in the magnitudes themselves, it was hypothesized that simulated magnitudes can be corrected using an independently estimated flow-duration curve. This hypothesis was evaluated by considering the performance of simulated streamflow observations and the performance of the relative timing of simulated streamflow. This evaluation was primarily focused on examination of errors in both the high and low tails of the streamflow distribution, defined as the lowest and highest 5% of streamflow, and considering changes in both bias and accuracy.

When observed flow-duration curves were used for bias correction, representing a case with minimal uncertainty in the independently estimated flow-duration curve, bias and accuracy of both tails were substantially improved and overall accuracy was noticeably improved. The use of regionally regressed flow-duration curves, which were observed to be approximately unbiased in the upper tails but were biased low in the lower tails, corrected the upper tail bias but failed to consistently correct the lower tail bias. Furthermore, the use of the regionally regressed flow-duration curves degraded the accuracy of the lower tails but had relatively little effect on the accuracy of the upper tails. Combining the bias-correction and accuracy results, the test with regionally regressed flow-duration curves can be said to have been successful with the upper tails (for which the regionally regressed flow-duration curves were unbiased) but unsuccessful with the lower tails. The effect on accuracy of the bias correction approach using estimated flow-duration curves was correlated with the accuracy with which each tail of the flow-duration curve was estimated by regional regression.

In conclusion, this approach to bias-correction has significant potential to improve the accuracy of streamflow simulations, though the potential is limited by how well the flow-duration curve can be reproduced. While conceived as a method of bias correction, this approach is an analog to a previously applied nonlinear spatial interpolation method using flow-duration curves to reproduce streamflow at ungauged basins. While using the nonexceedance probabilities of kriged streamflow simulations may improve on the use of single index streamgages to obtain nonexceedance probabilities, further improvements are limited by the ability to estimate the flow-duration curve more accurately.

*Code and data availability.* The data and scripts used to produce the results discussed herein can be found in Farmer et al. (2018).

*Competing interests.* No competing interests are present.

*Acknowledgements.* This research was supported by the U.S. Geological Survey's National Water Census. Any use of trade, product, or firm names is for descriptive purposes only and does not imply endorsement by the U.S. Government. We are very grateful for the comments of several reviewers, among whom was Dr. Benoit Hingray. The combined reviewer comments helped to greatly improve early versions of this manuscript.

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

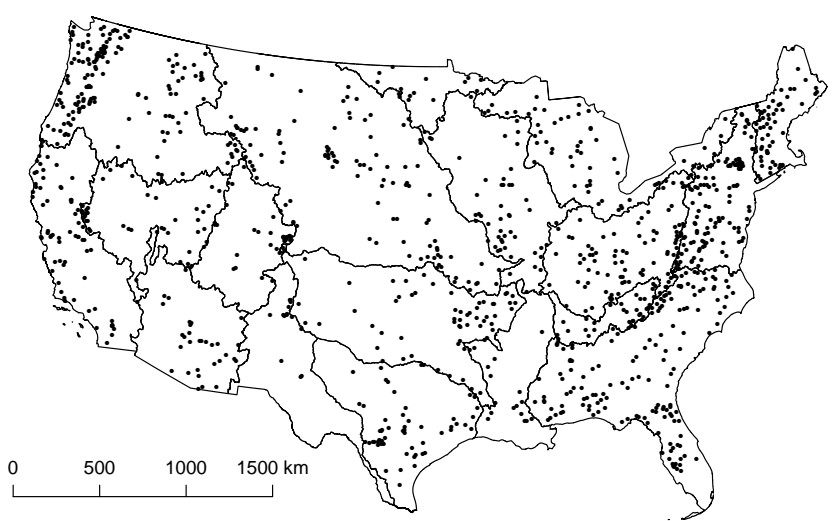

**Figure 1.** Map of the locations of 1168 reference quality streamgages from the GAGES-II database (Falcone, 2011) used for analysis. All streamgages used have more than 14 complete water years between 01 October 1980 and 30 September 2013. The outlines of 2-digit Hydrologic Units, which define the regions used here, are provided for further context.

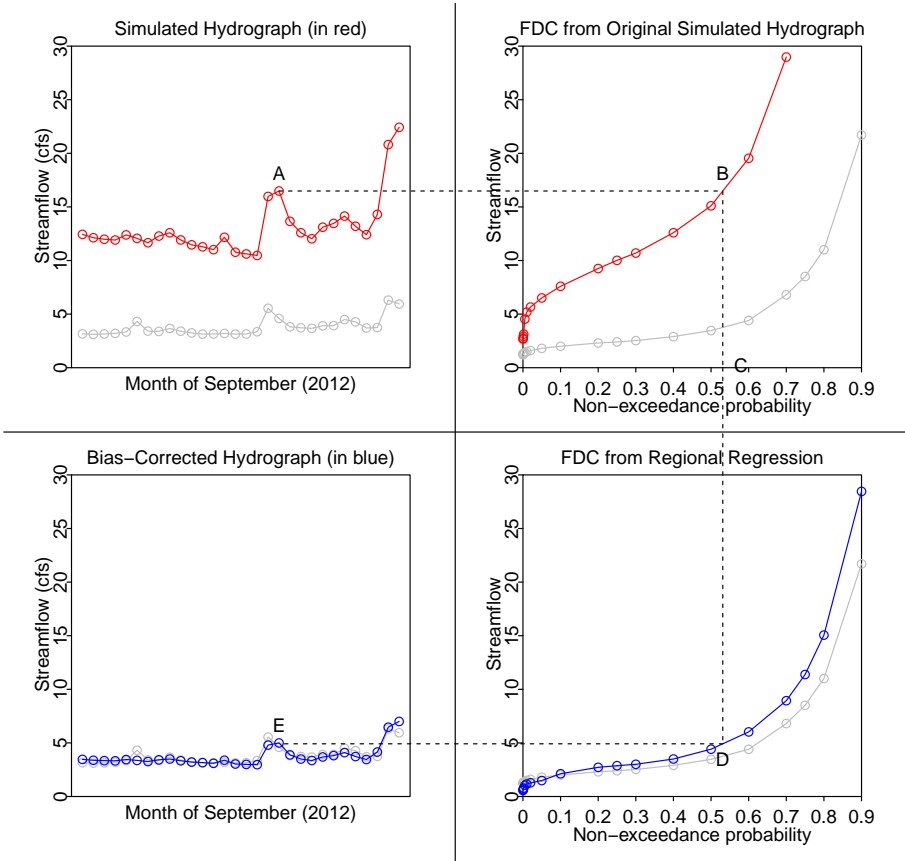

**Figure 2.** Diagram showing the bias-correction methodology applied here. The simulated daily hydrograph at the ungauged site is presented in the upper-left panel. For any particular point on the hydrograph (point A) the daily volume of streamflow can be mapped to a non-exceedance probability using the rank order of simulated streamflows (points B and C). With an independently estimated flow duration curve (FDC) from some procedure such as regional regression, the non-exceedance probability can be rescaled to a new volume (point D) and placed back in same sequence as the simulated streamflows (point E) to produce a bias-corrected hydrograph. This example is shown for one month, though the FDC applies across the entire period of record. As this data is based on an example site, the observed streamflows and FDC are shown in grey on each figure.

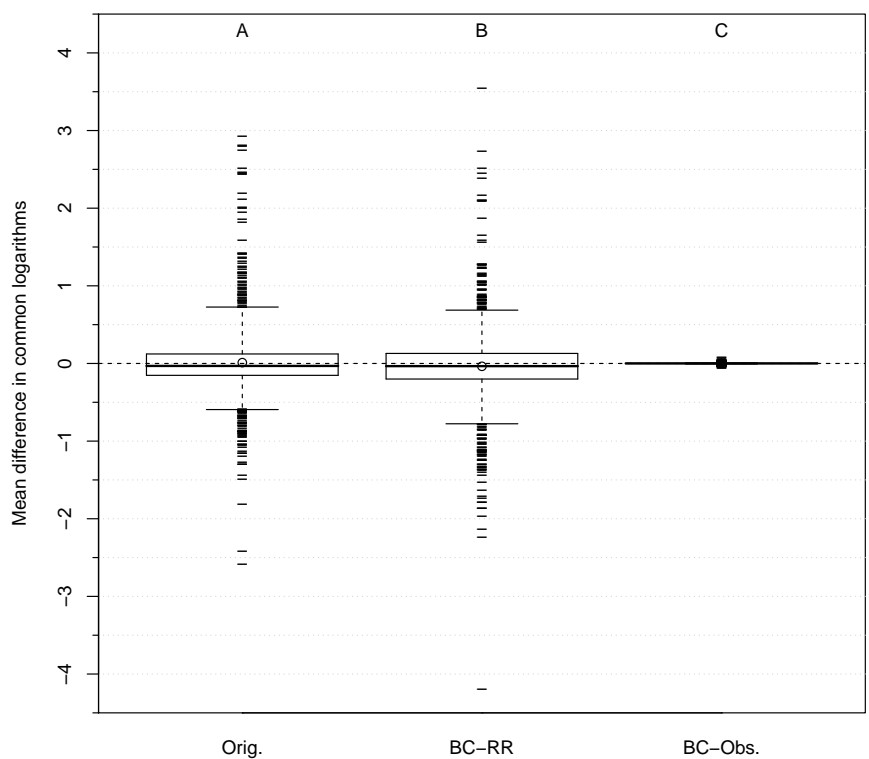

**Figure 3.** Distribution of logarithmic bias, measured as the mean difference between the common logarithms of simulated and observed streamflow (simulated minus observed) at 1168 streamgages across the conterminous United States. Orig. refers to the original simulation with pooled ordinary kriging, BC-RR refers to the Orig. hydrograph bias-corrected with regionally regressed flow-duration curves, and BC-Obs. refers to the Orig. hydrograph bias-corrected with observed flow- duration curves. The tails of the boxplots extend to the 5th and 95th percentiles of the distribution; the ends of the boxes represent the 25th and 75th percentiles of the distribution; the heavier line in the box represents the median of the distribution; the open circle represents the mean of the distribution; outliers beyond the 5th and 95th percentile are shown as horizontal dashes.

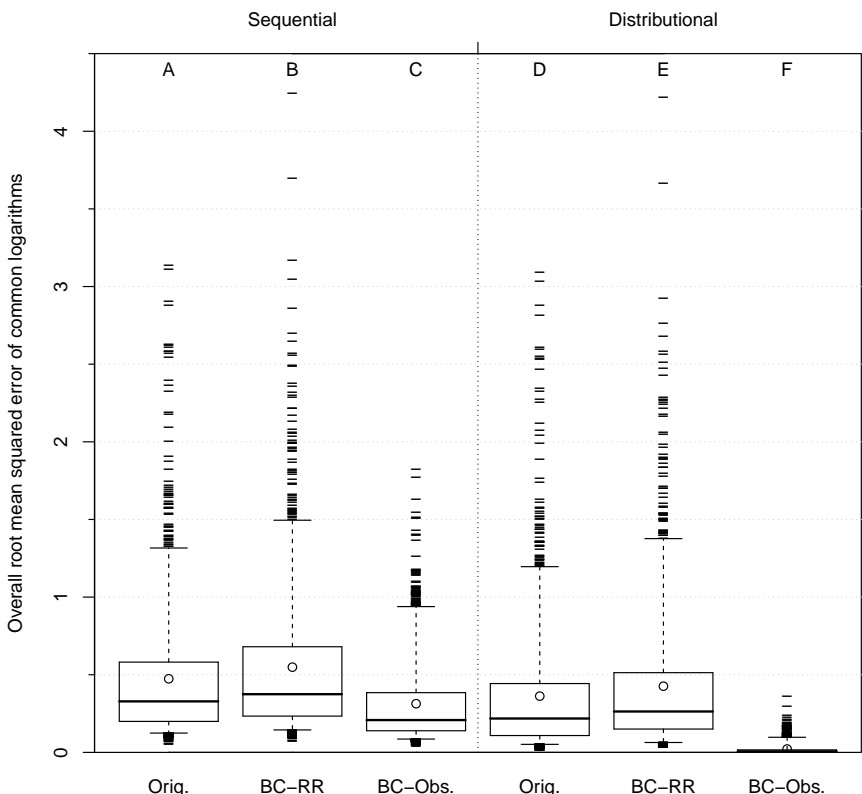

**Figure 4.** Distribution of logarithmic accuracy, measured as the root mean squared error between the common logarithms of observed and simulated streamflow at 1168 streamgages across the conterminous United States. Orig. refers to the original simulation with pooled ordinary kriging, BC-RR refers to the Orig. hydrograph bias-corrected with regionally regressed flow-duration curves, and BC-Obs. refers to the Orig. hydrograph bias-corrected with observed flow-duration curves. Sequential indicates that contemporary days were compared, while distributional indicates that days of equal rank were compared. The tails of the boxplots extend to the 5th and 95th percentiles of the distribution; the ends of the boxes represent the 25th and 75th percentiles of the distribution; the heavier line in the box represents the median of the distribution; the open circle represents the mean of the distribution; outliers beyond the 5th and 95th percentile are shown as horizontal dashes.

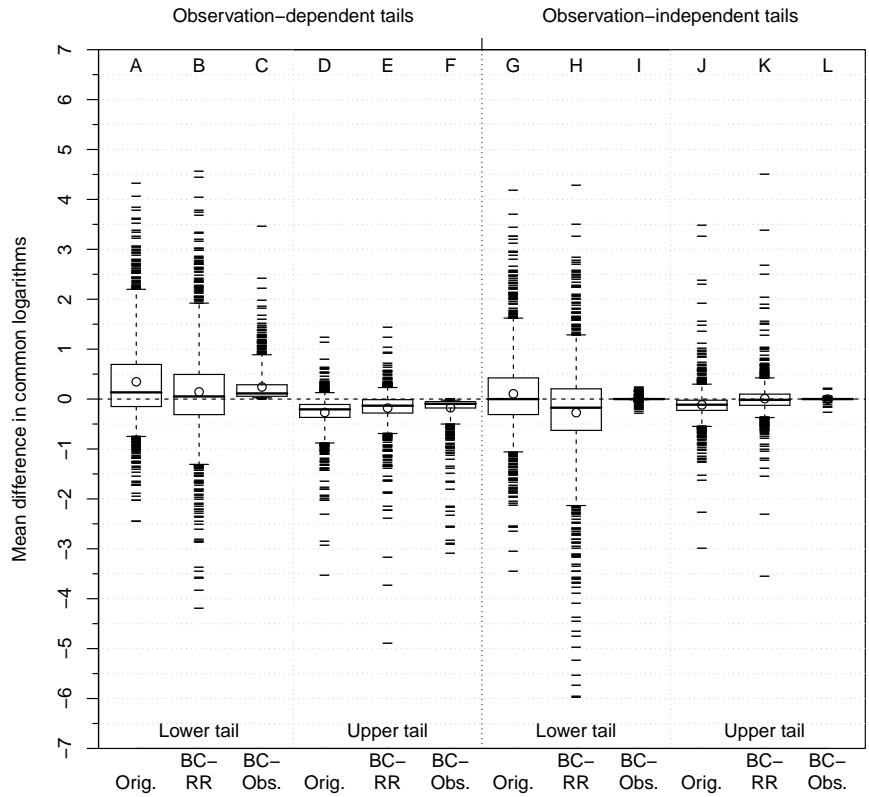

**Figure 5.** Distribution of logarithmic bias, measured as the mean difference between the common logarithms of simulated and observed streamflow at 1168 streamgages across the conterminous United States for observation-dependent and observation-independent upper and lower tails. Observation-dependent tails retain the ranks of observed streamflow, while matching simulations by day. Observation-independent tails rank observations and simulation independently. The upper tail considers the highest 5% of streamflow, while the lower tail considers the lowest 5% of streamflow. Orig. refers to the original simulation with pooled ordinary kriging, BC-RR refers to the Orig. hydrograph bias-corrected with regionally regressed flow-duration curves, and BC-Obs. refers to the Orig. hydrograph bias-corrected with observed flow-duration curves. The tails of the boxplots extend to the 5th and 95th percentiles of the distribution; the ends of the boxes represent the 25th and 75th percentiles of the distribution; the heavier line in the box represents the median of the distribution; the open circle represents the mean of the distribution; outliers beyond the 5th and 95th percentile are shown as horizontal dashes.

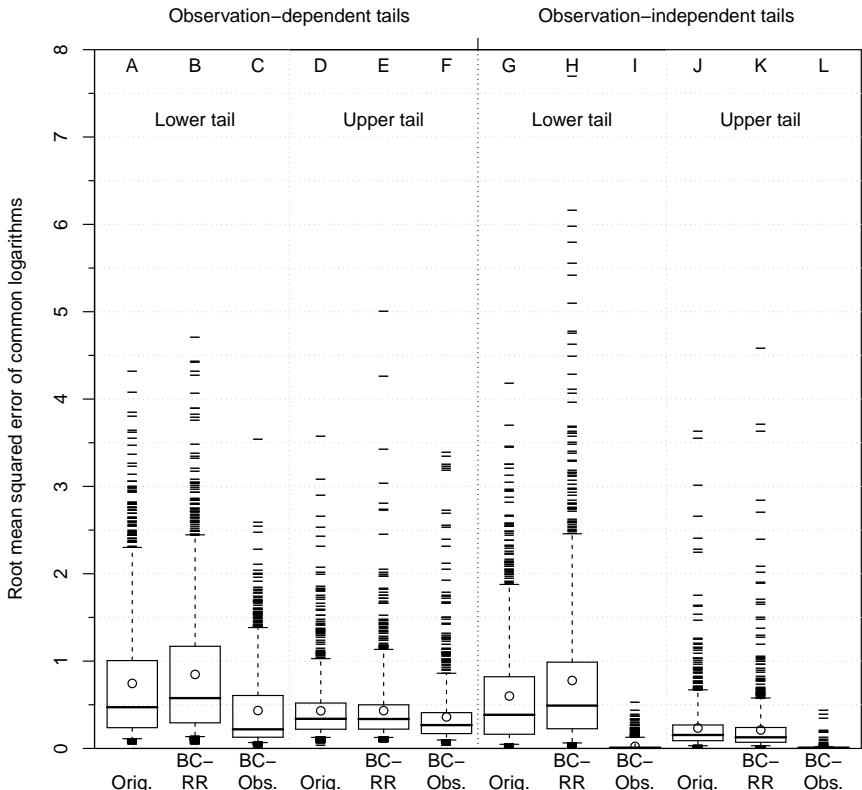

**Figure 6.** Distribution of logarithmic accuracy, measured as the root mean squared error between the common logarithms of simulated and observed streamflow (simulated minus observed) at 1168 streamgages across the conterminous United States for observation-dependent and observation-independent upper and lower tails. Observation-dependent tails retain the ranks of observed streamflow, while matching simulations by day. Observation-independent tails rank observations and simulation independently. The upper tail considers the highest 5% of streamflow, while the lower tail considers the lowest 5% of streamflow. Orig. refers to the original simulation with pooled ordinary kriging, BC-RR refers to the Orig. hydrograph bias-corrected with regionally regressed flow-duration curves, and BC-Obs. refers to the Orig. hydrograph bias-corrected with observed flow-duration curves. The tails of the boxplots extend to the 5th and 95th percentiles of the distribution; the ends of the boxes represent the 25th and 75th percentiles of the distribution; the heavier line in the box represents the median of the distribution; the open circle represents the mean of the distribution; outliers beyond the 5th and 95th percentile are shown as horizontal dashes.

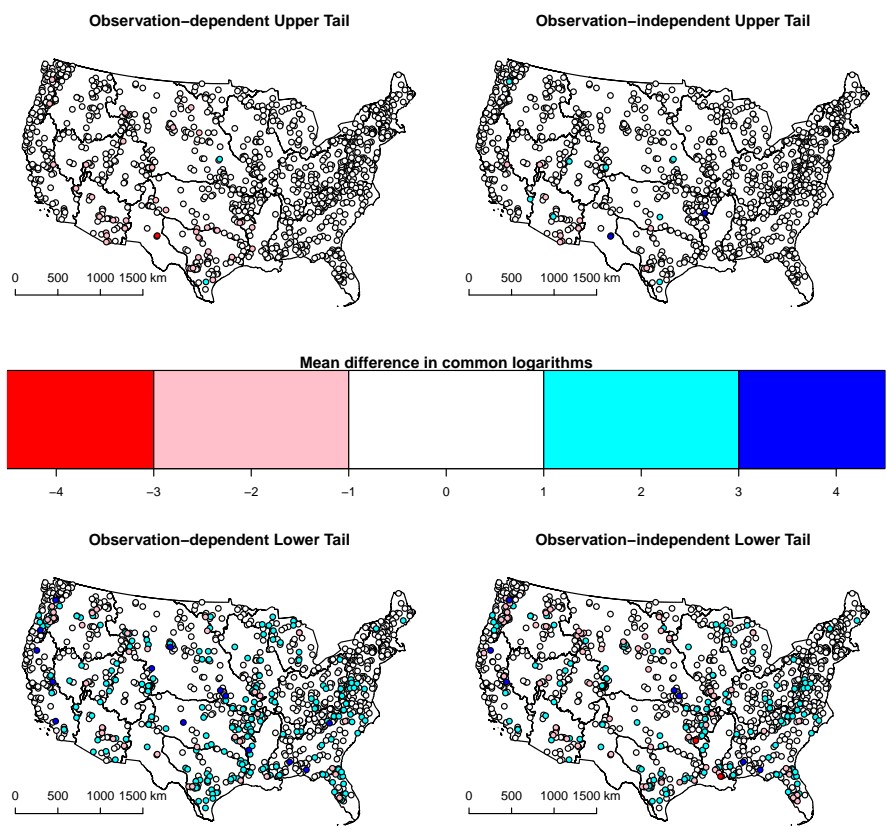

**Figure 7.** Maps showing the distribution of logarithmic bias, measured as the mean difference between the common logarithms of simulated and observed streamflow (simulated minus observed) at 1168 streamgages across the conterminous United States for observation-dependent and observation-independent upper and lower tails. Observation-dependent tails retain the ranks of observed streamflow, while matching simulations by day. Observation-independent tails rank observations and simulation independently. The upper tail considers the highest 5% of streamflow, while the lower tail considers the lowest 5% of streamflow. The bias is derived from the original simulation of daily streamflow using pooled ordinary kriging at 1168 sites regionalized by the 2-digit Hydrologic Units (polygons).

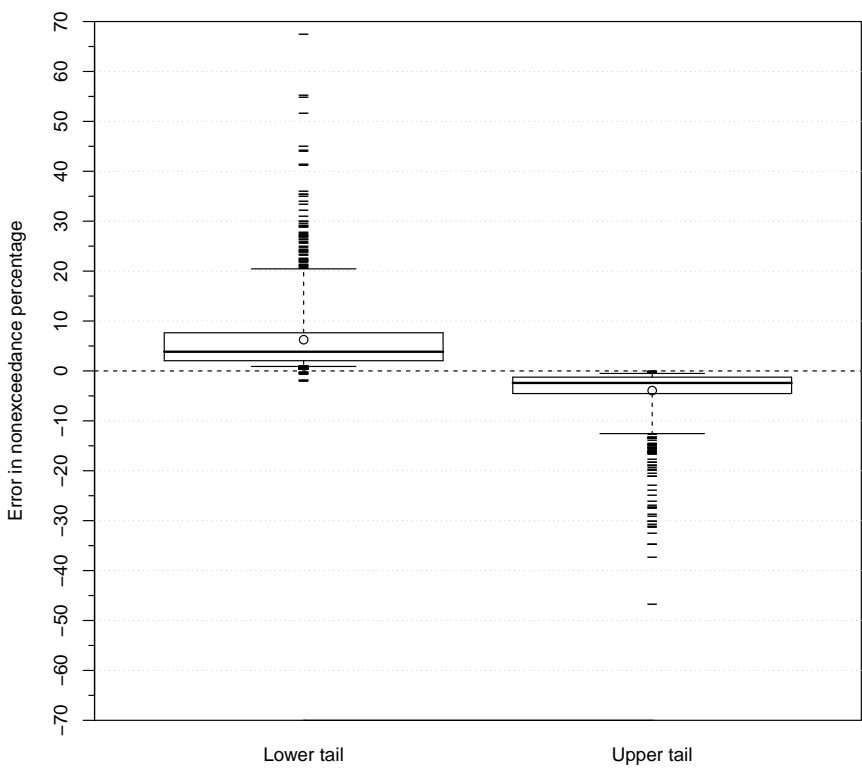

**Figure 8.** Distribution of mean error in the simulated nonexceedance probabilities of the lowest and highest 5% of observed daily stream-flow (simulated minus observed) at 1168 streamgages across the conterminous United States. The upper tail considers the highest 5% of streamflow, while the lower tail considers the lowest 5% of streamflow. The tails of the boxplots extend to the 5th and 95th percentiles of the distribution; the ends of the boxes represent the 25th and 75th percentiles of the distribution; the heavier line in the box represents the median of the distribution; the open circle represents the mean of the distribution; outliers beyond the 5th and 95th percentile are shown as horizontal dashes.

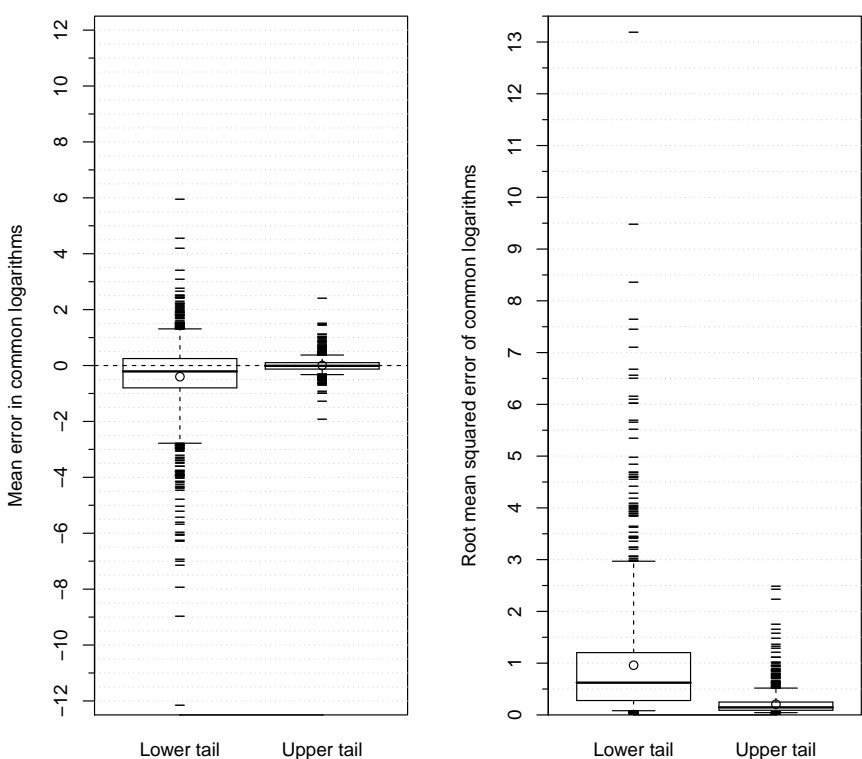

**Figure 9.** Distribution of logarithmic bias (left panel), measured as the mean difference between the common logarithms of quantiles of observed and simulated streamflow (simulated minus observed) at 1168 streamgages across the conterminous United States, and logarithmic accuracy (right panel), measured as the root mean squared error between the common logarithms of quantiles of observed and simulated streamflow at the same streamgage, in the upper and lower quantiles of regionally regressed flow-duration curves. The upper tail considers the 8 quantiles in the highest 5% of streamflow, while the lower tail considers the 8 quantiles in the lowest 5% of streamflow. The tails of the boxplots extend to the 5th and 95th percentiles of the distribution; the ends of the boxes represent the 25th and 75th percentiles of the distribution; the heavier line in the box represents the median of the distribution; the open circle represents the mean of the distribution; outliers beyond the 5th and 95th percentile are shown as horizontal dashes.

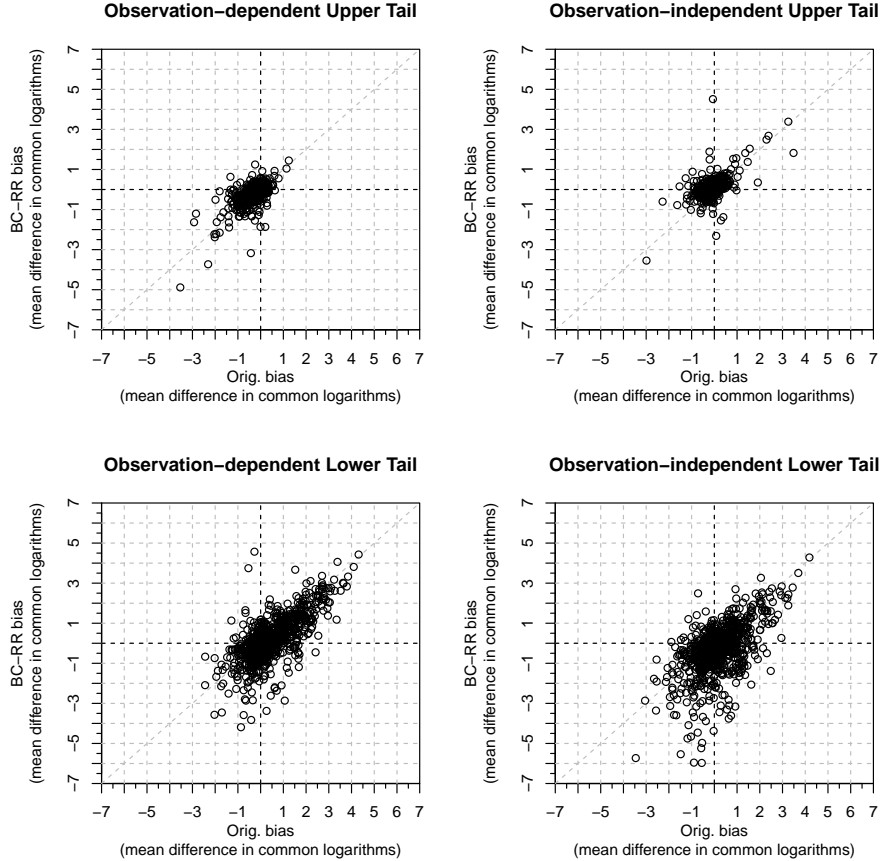

**Figure 10.** Scatter plots showing the correspondence of logarithmic bias, measured as the mean difference between the common logarithms of simulated and observed streamflow (simulated minus observed) at 1168 streamgages across the conterminous United States for observation-dependent and observation-independent upper and lower tails. Observation-dependent tails retain the ranks of observed streamflow, while matching simulations by day. Observation-independent tails rank observations and simulation independently. The upper tail considers the highest 5% of streamflow, while the lower tail considers the lowest 5% of streamflow. Orig. refers to the original simulation with pooled ordinary kriging, and BC-RR refers to the Orig. hydrograph bias-corrected with regionally regressed flow-duration curves.

**Table 1.** Measures of the distribution of logarithmic bias, computed as the mean difference between the common logarithms of simulated and observed streamflow (simulated minus observed) at 1168 streamgages across the conterminous United States for observation-dependent and observation-independent upper and lower tails. Orig. refers to the original simulation with pooled ordinary kriging, BC-RR refers to the Orig. hydrograph bias-corrected with regionally regressed flow-duration curves, and BC-Obs. refers to the Orig. hydrograph bias-corrected with observed flow-duration curves. Observation-dependent (OD) tails retain the ranks of observed streamflow, while matching simulations by day. Observation-independent (OI) tails rank observations and simulation independently. The upper tail observes the highest 5% of streamflow, while the lower tail considers the lowest 5% of streamflow. Significance is the p-value resulting from a Wilcoxon signed rank test with continuity correction, with the null hypothesis that the median of distribution is equal to zero and the alternative hypothesis that median is not equal to zero.

| | | Overall | | | | | Lower Tail | | | | | Upper Tail | | | | |
|---|---|---|---|---|---|---|---|---|---|---|---|---|---|---|---|---|
| | | Median | Mean | Interquartile Range | Standard Deviation | Significance | Median | Mean | Interquartile Range | Standard Deviation | Significance | Median | Mean | Interquartile Range | Standard Deviation | Significance |
| Orig | OD | -0.0318 | 0.0108 | 0.2752 | 0.4574 | 0.0067 | 0.1340 | 0.3469 | 0.8437 | 0.8918 | <0.0001 | -0.2060 | -0.2685 | 0.2590 | 0.3532 | <0.0001 |
| | OI | -0.0318 | 0.0108 | 0.2752 | 0.4574 | 0.0067 | -0.0007 | 0.1058 | 0.7347 | 0.8323 | 0.0245 | -0.1129 | -0.1165 | 0.2036 | 0.3420 | <0.0001 |
| BC-RR | OD | -0.0344 | -0.0364 | 0.3298 | 0.4992 | <0.0001 | 0.0539 | 0.1446 | 0.8040 | 0.9664 | <0.0001 | -0.1326 | -0.1808 | 0.2678 | 0.3827 | <0.0001 |
| | OI | -0.0344 | -0.0364 | 0.3298 | 0.4992 | <0.0001 | -0.1732 | -0.2723 | 0.8323 | 1.0893 | <0.0001 | -0.0162 | 0.0085 | 0.2240 | 0.3670 | 0.0547 |
| BC-Obs. | OD | 0.0004 | 0.0004 | 0.0017 | 0.0078 | <0.0001 | 0.1151 | 0.2426 | 0.2281 | 0.3225 | <0.0001 | -0.0957 | -0.1735 | 0.1284 | 0.2943 | <0.0001 |
| | OI | 0.0004 | 0.0004 | 0.0017 | 0.0078 | <0.0001 | 0.0000 | 0.0014 | 0.0035 | 0.0316 | 0.0018 | 0.0004 | 0.0009 | 0.0051 | 0.0144 | <0.0001 |
| Estimated FDC | | -0.0796 | -0.1270 | 0.4091 | 0.5525 | <0.0001 | -0.2101 | -0.3988 | 1.0485 | 1.3589 | <0.0001 | -0.0108 | 0.0047 | 0.2302 | 0.2611 | 0.1336 |

**Table 2.** Measures of the distribution of logarithmic accuracy, computed as the root mean squared error between the common logarithms of observed and simulated streamflow at 1168 streamgages across the conterminous United States for observation-dependent and observation-independent upper and lower tails. Orig. refers to the original simulation with pooled ordinary kriging, BC-RR refers to the Orig. hydrograph bias-corrected with regionally regressed flow-duration curves, and BC-Obs. refers to the Orig. hydrograph bias-corrected with observed flow-duration curves. Observation-dependent (OD) tails retain the ranks of observed streamflow, while matching simulations by day. Observation-independent (OI) tails rank observations and simulation independently. The upper tail observes the highest 5% of streamflow, while the lower tail considers the lowest 5% of streamflow.

| | | Overall | | | | Lower Tail | | | | Upper Tail | | | |
|---|---|---|---|---|---|---|---|---|---|---|---|---|---|
| | | Median | Mean | Interquartile Range | Standard Deviation | Median | Mean | Interquartile Range | Standard Deviation | Median | Mean | Interquartile Range | Standard Deviation |
| Orig | OD | 0.3286 | 0.4741 | 0.3818 | 0.4293 | 0.4722 | 0.7448 | 0.7649 | 0.7197 | 0.3394 | 0.4310 | 0.2998 | 0.3501 |
| | OI | 0.2182 | 0.3623 | 0.3347 | 0.4164 | 0.3852 | 0.6003 | 0.6583 | 0.6171 | 0.1542 | 0.2338 | 0.1800 | 0.2969 |
| BC-RR | OD | 0.3747 | 0.5489 | 0.4466 | 0.4827 | 0.5763 | 0.8476 | 0.8802 | 0.7633 | 0.3371 | 0.4331 | 0.2785 | 0.3913 |
| | OI | 0.2634 | 0.4264 | 0.3631 | 0.4609 | 0.4905 | 0.7780 | 0.7622 | 0.8626 | 0.1277 | 0.2116 | 0.1696 | 0.3209 |
| BC-Obs. | OD | 0.2080 | 0.3137 | 0.2454 | 0.2660 | 0.2186 | 0.4350 | 0.4789 | 0.4558 | 0.2674 | 0.3612 | 0.2400 | 0.3716 |
| | OI | 0.0066 | 0.0218 | 0.0115 | 0.0369 | 0.0066 | 0.0240 | 0.0096 | 0.0556 | 0.0084 | 0.0114 | 0.0050 | 0.0234 |
| Estimated FDC | | 0.4073 | 0.6220 | 0.5560 | 0.6699 | 0.6227 | 0.9600 | 0.9259 | 1.1495 | 0.1455 | 0.2068 | 0.1585 | 0.2207 |

**Table 3.** Measures of the distribution of changes in absolute logarithmic bias with bias correction, where absolute logarithmic bias is computed as the absolute value of the mean difference between the common logarithms of bias-corrected and simulated streamflow at 1168 streamgages across the conterminous United States for observation-dependent and observation-independent upper and lower tails, where the simulated streamflow was obtained with pooled ordinary kriging. BC-RR refers to the Orig. hydrograph bias-corrected with regionally regressed flow- duration curves, and BC-Obs. refers to the Orig. hydrograph bias-corrected with observed flow-duration curves. Observation-dependent (OD) tails retain the ranks of observed streamflow, while matching simulations by day. Observation-independent (OI) tails rank observations and simulation independently. The upper tail observes the highest 5% of streamflow, while the lower tail considers the lowest 5% of streamflow. Significance is the p-value resulting from a paired Wilcoxon signed rank test with continuity correction, with the null hypothesis that the median difference with respect to the original simulation is equal to zero, while the alternative hypothesis that median difference is not equal to zero.

| | | Overall | | | | | Lower Tail | | | | | Upper Tail | | | | |
|---|---|---|---|---|---|---|---|---|---|---|---|---|---|---|---|---|
| | | Median | Mean | Interquartile Range | Standard Deviation | Significance | Median | Mean | Interquartile Range | Standard Deviation | Significance | Median | Mean | Interquartile Range | Standard Deviation | Significance |
| BC-RR | OD | 0.0215 | 0.0385 | 0.2117 | 0.3274 | <0.0001 | 0.0163 | 0.0269 | 0.4866 | 0.5953 | 0.3710 | -0.0545 | -0.0526 | 0.1857 | 0.2690 | <0.0001 |
| | OI | 0.0215 | 0.0385 | 0.2117 | 0.3274 | <0.0001 | 0.0508 | 0.1588 | 0.5922 | 0.7885 | <0.0001 | -0.0273 | -0.0261 | 0.1946 | 0.2813 | <0.0001 |
| BC-Obs. | OD | -0.1382 | -0.2605 | 0.2280 | 0.3718 | <0.0001 | -0.1996 | -0.3859 | 0.5354 | 0.5572 | <0.0001 | -0.1111 | -0.1334 | 0.1551 | 0.3017 | <0.0001 |
| | OI | -0.1382 | -0.2605 | 0.2280 | 0.3718 | <0.0001 | -0.3492 | -0.5615 | 0.6255 | 0.6081 | <0.0001 | -0.1424 | -0.2160 | 0.1891 | 0.2821 | <0.0001 |

**Table 4.** Measures of the distribution of changes in logarithmic accuracy between original and bias-corrected simulations, where the logarithmic accuracy is computed as the root mean squared error between the common logarithms of bias-corrected and simulated streamflow at 1168 streamgages across the conterminous United States for observation-dependent and observation-independent upper and lower tails, where the simulated streamflow was obtained using pooled ordinary kriging. BC-RR refers to the Orig. hydrograph bias-corrected with regionally regressed flow-duration curves, and BC-Obs. refers to the Orig. hydrograph bias-corrected with observed flow- duration curves. Observation-dependent (OD) tails retain the ranks of observed streamflow, while matching simulations by day. Observation-independent (OI) tails rank observations and simulation independently. The upper tail observes the highest 5% of streamflow, while the lower tail considers the lowest 5% of streamflow. Significance is the p-value resulting from a paired Wilcoxon signed rank test with continuity correction, with the null hypothesis that the median difference with respect to the original simulation is equal to zero, while the alternative hypothesis that median difference is not equal to zero.

| | | Overall | | | | | Lower Tail | | | | | Upper Tail | | | | |
|---|---|---|---|---|---|---|---|---|---|---|---|---|---|---|---|---|
| | | Median | Mean | Interquartile Range | Standard Deviation | Significance | Median | Mean | Interquartile Range | Standard Deviation | Significance | Median | Mean | Interquartile Range | Standard Deviation | Significance |
| BC-RR | OD | 0.0331 | 0.0749 | 0.1636 | 0.2966 | <0.0001 | 0.0422 | 0.1028 | 0.3897 | 0.5377 | <0.0001 | -0.0019 | 0.0020 | 0.1111 | 0.2348 | 0.1532 |
| | OI | 0.0377 | 0.0641 | 0.2159 | 0.3294 | <0.0001 | 0.0601 | 0.1777 | 0.5646 | 0.7791 | <0.0001 | -0.0222 | -0.0222 | 0.1800 | 0.2751 | <0.0001 |
| BC-Obs. | OD | -0.0658 | -0.1604 | 0.1615 | 0.2794 | <0.0001 | -0.1554 | -0.3098 | 0.4079 | 0.4597 | <0.0001 | -0.0436 | -0.0698 | 0.1051 | 0.2621 | <0.0001 |
| | OI | -0.2056 | -0.3405 | 0.3138 | 0.3957 | <0.0001 | -0.3702 | -0.5763 | 0.6399 | 0.6027 | <0.0001 | -0.1450 | -0.2224 | 0.1805 | 0.2852 | <0.0001 |