# Peer review of "Bias correction of Simulated Historical Daily Streamflow at Ungauged Locations by Using Independently Estimated Flow-Duration Curves"

_Hydrology and Earth System Sciences, 2018_

## Referee Comment (RC1) · Anonymous Referee #1 · 18 Apr 2018

This is an interesting paper, which in my opinion could be made easier to follow with some adjustments. I make a few suggestions that I honestly believe could improve its readability and subsequent impact.

Major Remarks

1. My major problem is with the structure of the paper and with its illustrations: 1.1 Even a specialist of statistical hydrology could use one or two hydrographs (you only show us box plots!). An introductory graph with an example hydrograph and FDC could help

the reader understand your methodology. 1.2 I do not like the way you deal with all the aspects of the methodology in parallel, it makes things very difficult to understand what you are doing. I would have preferred a paper structure where (a) you show us what a "perfect" simulation of the FDC used for bias correction could give for results, then (b) you would show that due to the inherent uncertainty of FDC prediction at ungauged points you loose a lot of the theoretical advantage, while managing to improve overall bias 1.3 Last, I believe that in addition to box-plots, you should also show the reader some QQ plots to show that even if on average there is a reduction of bias, there will always be catchments where the bias correction method will increase the bias : e.g. a plot showing the original low flow bias vs the bias corrected low-flow bias (with one point per catchment), and then the same for high flow.

Minor Remark 2. P2L15. "the nature of this approach..." : I have difficulties to understand this sentence...

---

## Referee Comment (RC2) · Anonymous Referee #2 · 30 Apr 2018

General comments

The work presents a method that aims to produce unbiased time series of daily discharges for ungauged basins. It consists in correcting the distributional bias in the time series of simulated discharges obtained for this ungauged basin from some simulation model (in the present case, the simulation model is a regression model where the daily discharge of each day is estimated via kriging from gauged stations in the neighborhood of the target ungauged station). The correction is a quantile –mapping correction where the reference statistical distribution is obtained from the Flow-Duration-Curve of

the target location, estimated via regionalization again from the observed FDC estimated for the neighboring stations.

The issue is a very relevant one and the approach a promising one. The novelty of the approach with respect to other similar studies (mentioned in the manuscript) has to be clarified.

More specifically, the aim of the paper is not well explained and it has taken me more than 2 hours to understand what was done in the paper. One or possibly several graphical schemes are definitively required to figure out the methodology / objective of the paper and data used for the application of the method. A reformulation of the introduction and of the abstract is for me also necessary. Showing figures with hydrographs or FDC is also required (boxplots are not informative enough).

The formulation in the abstract for instance is somehow clumsy : "Based on an existing approach that separates the simulated streamflow into components of timing and magnitude, the timing component is converted into simulated nonexceedance probabilities and rescaled to new volumes using an independently estimated flow-duration curve (FDC)." I think that the authors do not rescale the timing component. They just scale each daily discharge of the time series with a multiplicative correction factor that depends on the non exceedance probability of that discharge... The correction function is estimated from the FDC of the target location, estimated from the observed FDC of gauged stations in the region. I would have understood instantaneously the objective / approach of the work that way (I am perhaps wrong, but I guess this is roughly what you do). This has definitively to be clarified (here and everywhere else in the manuscript).

It is also not clear that the authors want to estimate unbiased time series of discharge at ungauged sites. This has to be clearly stated. One thus needs to have a regional simulation model and a regional model for the FDC estimation. The interest of the configuration where an observed FDC is used to bias correct the simulated time series has to be made clear in this context. This approach is presented as an alternative

estimation approach. It is actually not and this is really confusing.

The results of the study were rather expected. The authors find that the combined "time series simulation"+"FDC bias correction approach" is unable to correct the bias in the simulated hydrograph. The authors explain this is due to the bad quality of the regionalized FDC which present a rather significant distributional bias for all discharge percentiles. By construction, the "poor" result of the FDC bias correction was thus expected. The interest of the method would have been for a configuration where the distributional bias obtained from the "time series simulation" approach is larger than that of the regionalized FDC. Obviously, this is unfortunately here not the case. At least in average for the 1168 basins considered in the work. More convincing results could have perhaps been presented considering different configurations of basins with different relative performances of the two approaches considered here for the combination. We would have seen that the method works in configurations where reginoalised FDC are good.

Focused Remarks :

P2. "As attested to by many researchers focused on the reproduction of historical streamflow, this bias commonly appears as a general overestimation of low streamflow and underestimation of high streamflow". This statement is not true. Simulations with any hydrological model will lead to the overestimation of some high stream flow and the underestimation of the others. The same for low flows : some are over-, some are underestimated. In rainfall-runoff models, it clearly depends on the amount of rainfall estimated from the small number of raingauge available, and frequently not numerous enough to have the right estimate of rainfall input. It also depends on the quality of the model which is limited for the simulation of specific hydro processes. The fact that a given model systematically overestimate lowflows and underestimate highflows surely means that the model is not good enough and could likely be improved. As a matter of fact, some models present those limitations and this justify the present work. But a general statement can not be given there on such a case. (Note that this statement is

invalidated in your work by the results and comments you mention (p6 – line 20/25). Please reformulate.

P2. Ln 22. The difference / novelty of the present methodology with that of Fennessey (1994) and Hughes and Smakhtin (1996) is not clear. Please clarify.

P3. The description of the content of the paper is missing at the end of the introduction.

P3. Simulated discharges. Âń initial predictions of daily streamflow values for each streamgage were obtained by applying the pooled ordinary kriging approach (Farmer, 2016) to each 2-digit Hydrologic Unit". The approach has to be better explained. It is not possible to understand how time series are obtained for any given gauge from what is said in the paragraph ("It builds a single, time-invariant semivariogram model of cross-correlation that is then used to estimate ungauged streamflow as a weighted summation of all contemporary observations"). A summary methodological scheme could be helpful. Is the variogram model estimated independently for each day ? for each target location from stations in its neighborhood (or do you use all stations of the United States to estimate the daily discharge of a given station) ? What is a 2-digit Hydrologic Unit ? Does the leave-one out procedure applies for the target location ?

P4. Ln 1-8 : Please clarify. "Daily FDCs were developed independently of the stream-flow simulation procedure by following a regionalization procedure similar to that of Farmer et al. (2014). Âż You mention "These same percentiles were then estimated using a leave-one-out cross-validation of regional regression". The objective/process of this estimation was not clear. Please make clear that the FDC used to bias correct your target station is obtained from a regional FDC model, obtained from all (or a part of) the stations close to this target station.

P4. Explanatory variables. Please comment the possible correlations between those. Have you looked / used for uncorrelated sets of explanatory variables ?

P4 – ln 10 : what are "best-subsets regression". Best sub-sets of what ? What is the

difference with "three contiguous streamflow regimes Âż you mention in line 17.

P4; Please clarify. "the percentiles were grouped into a maximum of three contiguous streamflow regimes based on the behavior of the unit FDCs in the region". What is a unit FDC ? How are defined the regimes ? Are they relative to different parts of the FDCs or to different sets of different basins within the region ? Do you group 95th and 90th percentiles for instance ?

P4. "The percentiles in each regime were estimated by the same explanatory variables, allowing only the fitted coefficients to change.." Do you mean that a regression is estimated for each of the Twenty-seven percentiles you considered ? If yes, a very significant risk on a non monotonic behavior of the FDC is likely. Why not working with a analytical model fitted to the FDC (and then propose a regionalization of the parameters of this model ?). This issue should be discussed somewhere.

P4. "Further details on the approach of percentile grouping this methodology can be explored in the associated data and model archive Farmer et al. (2018)." The given reference is a technical report. The soundness of the ""percentile grouping approach" is thus uncertain. It has thus to be fully explained in the present manuscript or it has to be described in another a peer reviewed journal.

P4. The section on the bias correction approach has to be made clear. Especially how the bias correction works and with which data in the case of ungauged catchments. Again a synthetic graphical scheme of the BC approach would be usefull. • You say "The nonexceedance probabilities were then converted to standard normal quantiles and linearly interpolated along two types of independently 30 estimated FDCs: the regionally regressed FDCs and the observed FDCs determined by applying the Weibull plotting position." It is something to do with quantile mapping ? • I can imagine that a correction function can be established for any given station where you have both observations and simulations. Is the idea here to transfer the correction function obtained for a gauged catchment to neighboring ungauged ones ? If yes, how do you

define the neighborhood ? how do you consider the different corrections functions you can obtained for the different gauges stations you may have in the neighborhood of your target location ?

P4. Ln – 29/30 : "The nonexceedance probabilities were then converted to standard normal quantiles and linearly interpolated along two types of independently estimated FDCs: the regionally regressed FDCs and the observed FDCs determined by applying the Weibull plotting position." » This is confusing. In the ungauged basin configuration : only one FDC is expected to be used : the regionalized one. The observed FDC is not expected to be available in a ungauged catchment. You use it here only to estimate the added value of the quality of the regionalized FDC on the resulting bias corrected discarhge time series.The "independently estimated FDC" you mention in line 31 (and basically everywhere else in your manuscript) should first refer, I guess, to the regionalized FDC (and not to the FDC from observations).

P5. Ln 3: reformulate : "by correcting the simulated volumes to an independently estimated FDC. Âż

P5. Evaluation :

Ln 10- the two different evaluations approaches were not clear for me at first. A reformulation would be worth Ln 20 & follow.  c Evaluation for the whole FDC, or for a given tail : what is the evaluation crietiron : the mean of the bias estimated between each pair of percentiles ? the biais between the mean of the percentiles for the raw and corrected data respectively ??  c Evaluation on observation-independent tails and observation dependant tails. A graphical scheme please to explain what is done in the second case, at least in a supplementary material ! Results : all results are given in the form of boxplots. This is likely not enough to understand how the methodology work and how good it is. To give for a selection of stations the different curves (observed / regressed / regressed+bias corrected) would be helpful (with a good performance station and a bad one for instance) P5 ln33 and p7 . Ln 25. I find the term "timing"

and "error in timing" not appropriate. You could perhaps say " an error in timing of the percentiles". This is however more an error in the temporal structure of your simulated time series. This results from an error in volume which is one day an over-estimation of the true volume and the day after an under-estimation.

P7 ln34. You say "These timing errors also almost result in errors in a particular direction: low for high flow and high for low flows". You have perhaps such a mean behavior but as mentioned above, you have a number of low flows that are overestimated but also a number of low flows (less frequent) that are underestimated... Please reformulate to put this statement in perspective.

P6. Ln 2 : "Figure 4 and Table 1 summarize the tail bias in all approaches to streamflow simulation considered here.". What are the 3 different approaches ? This has to be clearly explained previously. The BC-Obs is not an "approach" similar to the 2 others as you do not know in principle the observed discharge for the target ungauged basin. It just allows you to identify the influence of the quality of the regionalised FDC. A reformulation is required when relevant in all the manuscript. The presentation of the method has probably also to be restructered to make it clearer.

P7. 1. The analysis of the second paragraph in this section 3.1 is clumsy. The Observed FDC is in principle perfect and thus the bias in simulation for the observation independent evaluation case should fully vanish after correction. You should have nothing or roughly nothing to comment here. The results should be perfect. Why is there some remaining bias with the BC-obs approach ?? Please comment. (could it be a difference in the time period used for the simulation and the time period used to identifiy the observed FDC? Is it something related to the epsilon value you add to discharge data for the logarithmic transformation issue ? to the reduced number of percentile used to describe the FDC ? something else ? ) The comment on this in the conclusion section has also to be modified accordingly (p12 – ln 1/10). P7. 24: "To understand the effect of errors in timing further, consider Figure 6, which shows the mean error in the nonexceedance probabilities of the observation-dependent upper and lower tails."

I can not understand (just guess) what is refered to here ? Please clarify.

Fig 2 and following : how many data are used for each boxplot (#stations x #percentiles ???). Is there one point for each station/percentile ?

Discussion : what about the likely seasonal dependence of the correction function ? Is there some potential for improvement here ? The estimated FDCs are composed of 27 quantiles, of which the upper and lower tails contain only the eight values with nonexceedance probabilities 95% and larger and 5% and smaller, respectively.). A comment on the number of quantiles used to describe the FDC would be worth (a sensitivity analysis of the results to this number could be also included in a discussion section)

What is the influence of the duration of the observation time period used to estimate the observed FDC > on the quality of the FDC estimation and then on the quality of the bias correction ?

Is there any dependence of the results to the hydroclimatic context of the basin ? How is it structured is space in US ?

Minor remark : P2. Ln10. The interest of the "long term forecast term" here is not clear. It seems to be out of the scope of the work. To be better explained. What is long term ?

P2. 15/20 : » this paragraph is not clear »> to be clarified / double-formulate. The "interpolation of non-exceedance probabilities along the FDC" is a rather clumsy formulation. What does it mean ?

p2. Ln 33 : please clarify what is meant there P2. Ln 35 and following. This does not belong to the introduction but to a discussion section. The discussion should probably give the evaluation of the other method suggested here (Additional research to explore if estimating nonexceedance probabilities directly, as opposed to the conversion of simulated streamflow to nonexceedance probabilities used here, might further improve

nonlinear spatial interpolation using FDCs or simulation more generally.

p3. Ln 3 : give the structure of the paper

p5. "The root-mean-squared error of the common logarithms of streamflow and the differences therein were used to quantify accuracy." Do you mix streamflow and differences between streamflow in the computation of a single RMSE criterion ? If yes, I fear it is not relevant or please clarify / justify.

P6. Ln 9 – add a subsection title "simulated hydrographs without correction"

P6 – 25. "These results show upward bias in lower tails and downward bias in upper tails." No, this is not the case in general. See your paragraph above.

P9 "For the observation-independent case, the errors are removed almost completely, and the remaining errors in the observation-dependent case mimic the timing (nonexceedance probability) errors." This is not true (only if the observed FDC is used)

P10. 2 : "The change in the absolute bias of the observation-independent lower tail has a 0.72 Pearson correlation with the absolute bias of the lowest eight percentiles of the FDC estimated with regional "regression." I do not understand what is meant here.

Figures P10- 6 "as regional regression is not the only tool for estimating FDCs, improved methods for DC estimation would further increase the impact of this bias-correction procedure." Mention other such methods.

P10.15 "It may not always be possible to determine the accuracy with which a given FDC estimation technique might perform, making it difficult to determine if these results can be generalized." There is no reason why the accuracy of any FDC regionalization approach could not be assessed (this is done in all FDC regionalization study). At least with a leave-one out procedure. Before applied for the bias correction of any time series simulation model, this quality should be checked and estimated to be better than that of the simulation model.

Fig. 1 – What is meant here : "The outlines of 2-digit Hydrologic Units are provided for further context. Fig. 2 : BC-RR and BCObs have to be defined in the main text. Figures and tables : simplify the captions : a number of repetitions could be removed (and a reference to the caption details of one reference figure added in the caption of all other figures) Are the 3 tables usefull ?

———————————————————

---

## Referee Comment (RC3) · Anonymous Referee #3 · 15 May 2018

General comments:

The authors presented a bias-correction procedure useful for improving the accuracy of simulated daily streamflow series by using independently estimated flow-duration curves (FDCs). Although the procedure itself is not completely new (references of previous studies are included in the manuscript), the study is interesting as it considers an extended database in the US and focuses on the reproducibility of upper and lower tails, distinguishing between observation-dependent and observation-independent tails. This aspect is meaningful for highlighting the effect of timing on distributional bias. The study concludes that the significant potential of the bias-correction procedure is limited by the accuracy of the FDCs estimation method.

The paper is well structured and written, but some changes should be applied for making it more readable and for emphasizing some important aspects. I believe it is suitable for publication in HESS after the authors address some issues reported in the comments below.

Specific comments:

[1] I would suggest to explain more in details the "Bias correction" procedure at Section 2.3. For instance, at Page 4 Lines 29-30, the sentence "[...] linearly interpolated along two types of independently estimated FDCs" is rather ambiguous: I found difficult to understand whether the authors refer to the resampling of the curves, or perhaps to the prediction of FDC quantiles, which is carried out with a linear regression. I would recommend to rephrase this sentence and add more information to it, in order to clarify better this fundamental aspect of the proposed procedure. Moreover, at least one figure could be useful for clarifying the procedure. I can suggest to show at least two plots, where the authors may report standard normal quantiles vs. logarithmically transformed streamflow percentiles for, in turn, (a) regionally regressed FDC and pooled ordinary kriging curve, and (b) observed FDC and pooled ordinary kriging curve. Finally, in my opinion, the bias-correction section should be extended: I recommend to add more detailed information about how the bias correction is applied to the simulated streamflows, maybe introducing a figure vignette, or, likewise, describing the procedure point by point.

[2] I would stress more that in the majority of possible practical applications of the proposed method (i.e. predictions in ungauged sites), using observed FDCs for the bias correction would not be possible. Indeed, the only exception could be represented by those catchments in which we want to simulate streamflows for a given period, even though we have streamflow data for another period. I would add this reasonings to the

revised version of the paper.

[3] Page 10, Lines 11-13 – The authors highlight that "initial exploration did not find a strong regional component to performance of the bias correction method". In order to better support this sentence and to improve the effectiveness and completeness of the study, the authors could better discuss the spatial distribution of performance, especially given the high climatic variability among the conterminous United States. Therefore, I would suggest to add and discuss a new figure (or figures; e.g. a set of maps, similar to Figure 1), showing the spatial distribution of bias and root mean squared error in the study region for at least a couple of the cases considered in the study (Orig., BC-RR, BC-Obs.; upper tail, lower tail; observation-dependent tail, observation-independent tail; sequential, distributional, etc.).

Technical comments:

[1] In Figure 1, the differences between not-selected and selected streamgauges are not clear: in some areas, crosses overlap with points and the distinction is not simple. Maybe using different colors and symbol sizes might highlight better the differences between these two categories.

[2] Page 3, Line 24 – It is not clear what a 2-digit Hydrologic Unit is. Could you please explain?

[3] Page 3, Line 16 (and elsewhere) - "cubic feet per seconds (cfs)" is used. I believe that the International System of Units should be used in scientific papers. At the same time, if cfs is the standard adopted by USGS (and in the GAGES-II database), I think that at least the conversion factor to $m^3\ s^{-1}$ should be indicated in parentheses the first time that cfs is mentioned.

[4] Page 4, Line 3 - It seems that you are referring to period-of-record FDCs; am I correct? Could you please state that explicitly?

[5] Page 4, Line 10 - It is not clear to me what "best-subsets regression" is. Could you

please clarify and/or add at least one reference?

[6] Page 4, Lines 13-15 and 20-22; Page 8, Lines 19-21 - I would suggest to remove parentheses.

[7] Page 5, Lines 9-10 – I would recommend to add an equation showing the expression "ten to the power of the difference and subtracting one from this quantity". I would also suggest to explain why you are referring to this equation for computing the percentage. Moreover, the authors could show equations for bias and root mean squared error, respectively, when introducing them.

[8] Page 5, Line 10 – The authors write "root-mean-squared error", while use "root mean squared error" in the remainder of the text. I would suggest to use the same expression everywhere ("root mean squared error" should be fine).

[9] Page 5, Line 13 – Could you please add a reference for the Wilcoxon signed rank test?

[10] In Tables' captions, the acronyms "OD" and "OI" are used instead of "observation-dependent" and "observation-independent", respectively. You could use these acronyms also in the body of the text, in order to improve the readability; e.g. "OD-tail" instead of "observation-dependent tail" and "OI-tail" instead of "observation-independent tail".

[11] Page 10, Lines 6-7 – Could you please report some other methods (please provide references) for estimating FDCs?

[12] Page 11, Line 16 - Please delete "Summary and conclusions".

[13] Captions of Figures 2, 3, 4, 5 – Please remove comma in "pooled, ordinary kriging".

[14] In the body of text, there are some references to a recent study by Pugliese et al. (2017), which is currently under review. I would suggest to update these references after its possible acceptance/publication.

---

## Author Response (AR1)

At the outset, we wish to thank our reviewers and our handling editor for their deep consideration of our work. Their comments have provided a substantial improvement to the manuscript. The following contains all the original comments, the original author responses and final author responses directing the reader to the relevant changes. All line numbers in author comments refer to the tracked-changes documents included here.

**5 1 First Anonymous Reviewer**

**Reviewer Comment 1:** This is an interesting paper, which in my opinion could be made easier to follow with some adjustments. I make a few suggestions that I honestly believe could improve its readability and subsequent impact.

Author Response 1: Thank you for taking the time to review our manuscript. There is certainly a lot of material presented, so we appreciate your recommendations for improving readability. If you continue to have concerns, do not hesitate to comment again.

**Final Author Response 1:** We wish to thank you again for your consideration of this work. Your suggestions, particularly the figures, have allowed us to better present our material and addressed some of the concerns raised by other reviewers.

**Reviewer Comment 2:** Even a specialist of statistical hydrology could use one or two hydrographs (you only show us box plots!). An introductory graph with an example hydrograph and FDC could help the reader understand your methodology.

15 Author Response 2: For the revision, based on this and other reviews, we will include a two-panel graphic showing the overlay of an observed and simulated hydrograph (for a representative site) with a second panel showing the overlay of observed and simulated flow duration curves. We will also consider a figure that shows the steps of the methodology.

Final Author Response 2: In the end, we felt that a single graphic of the methodology provided all the context needed without providing too many figures. This additional figure is presented and discussed on page 6, line 17 of the tracked-changes
document. With this figure, the reader can easily see how an example hydrograph would be bias corrected.

**Reviewer Comment 3:** I do not like the way you deal with all the aspects of the methodology in parallel, it makes things very difficult to understand what you are doing. I would have preferred a paper structure where (a) you show us what a "perfect" simulation of the FDC used for bias correction could give for results, then (b) you would show that due to the inherent uncertainty of FDC prediction at ungauged points you loose a lot of the theoretical advantage, while managing to improve overall bias

**25 overall bias**

40

10

Author Response 3: This is a great suggestion for the flow of the paper. Other reviewers also made suggestions about the flow of the paper. We definitely plan to make revisions for clarity but need to determine which ideas are most appropriate. Final Author Response 3: The structure proposed is used in the Results section of the paper. With the inclusion of our new

Final Author Response 3: The structure proposed is used in the Results section of the paper. With the inclusion of our new methodology figure (page 6, line 17), the methodology is much more accessible.

30 **Reviewer Comment 4:** Last, I believe that in addition to box-plots, you should also show the reader some QQ plots to show that even if on average there is a reduction of bias, there will always be catchments where the bias correction method will increase the bias : e.g. a plot showing the original low flow bias vs the bias corrected low-flow bias (with one point per catchment), and then the same for high flow.

Author Response 4: This will be a useful figure, so we will consider weaving it into our manuscript.

**Final Author Response 4:** This new figure is presented and discussed on line 27 of page 13. This excellent addition allows us to highlight the variability of performance across sites and across tails.

**Reviewer Comment 5: P2 L15. "the nature of this approach. . . " : I have difficulties to understand this sentence. . .**

Author Response 5: This sentence is meant to point out that the timing of the raw simulation is not altered, only the magnitude. That is, the methodology assumes that the timing (sequence of relative rankings) is "good". We will find a better way to say this. One possible revision might be, "This approach assumes that, while the streamflow magnitudes of a historical simulation are biased, the timing or rank-order of the streamflows are relatively accurate."

Final Author Response 5: This revision appears on line 19 of page 2.

**2 Second Anonymous Reviewer**

**Reviewer Comment 1:** The work presents a method that aims to produce unbiased time series of daily discharges for ungauged basins. It consists in correcting the distributional bias in the time series of simulated discharges obtained for this ungauged basin from some simulation model (in the present case, the simulation model is a regression model where the daily discharge

5 of each day is estimated via kriging from gauged stations in the neighborhood of the target ungauged station). The correction is a quantile –mapping correction where the reference statistical distribution is obtained from the Flow-Duration-Curve of the target location, estimated via regionalization again from the observed FDC estimated for the neighboring stations.

The issue is a very relevant one and the approach a promising one. The novelty of the approach with respect to other similar studies (mentioned in the manuscript) has to be clarified.

10 Author Response 1: Thank you for your deep consideration of our manuscript. Your comments have helped us to identify weaknesses in presentation that substantially improve our presentation.

**Final Author Response 1:** Thank you, again, for your attention to our work. Addressing your comments surely helped us sharpen the presentation of our analyses.

Reviewer Comment 2: More specifically, the aim of the paper is not well explained and it has taken me more than 2 hours to
understand what was done in the paper. One or possibly several graphical schemes are definitively required to figure out the methodology / objective of the paper and data used for the application of the method. A reformulation of the introduction and of the abstract is for me also necessary. Showing figures with hydrographs or FDC is also required (boxplots are not informative enough).

- Author Response 2: Based on your comments and those of another reviewer, we are proposing to reorganize the manuscript and present several additional graphics. One will be a two-panel figure where one panel overlays observed and simulated hydrographs, while the second panel shows observed and simulated flow duration curves. A second figure will show how the methodology proceeds from simulated hydrograph, through estimated FDC to bias-corrected hydrograph. We will also take another look at the clarity of the introduction and the abstract.
- Final Author Response 2: On page 2, line 2 of the tracked-changes document we have added an explicit statement of the
  objective and novelty of this work. Novelty is further discussed on page 6, line 24. In addition, we have added the figure discussed on page 6, line 17, to represent our methodology. This figure allows the reader to better understand both our objective and our methodology.

**Reviewer Comment 3:** The formulation in the abstract for instance is somehow clumsy : "Based on an existing approach that separates the simulated streamflow into components of timing and magnitude, the timing component is converted into

- 30 simulated nonexceedance probabilities and rescaled to new volumes using an independently estimated flow-duration curve (FDC)." I think that the authors do not rescale the timing component. They just scale each daily discharge of the time series with a multiplicative correction factor that depends on the non exceedance probability of that discharge. . . The correction function is estimated from the FDC of the target location, estimated from the observed FDC of gauged stations in the region. I would have understood instantaneously the objective / approach of the work that way (I am perhaps wrong, but I guess this is roughly what you do). This has definitively to be clarified (here and everywhere else in the manuscript).
- Author Response 3: The reviewer is correct, and we propose revising this sentence to "Based on an existing approach that separates the simulated streamflow into components of timing and magnitude, the timing component is converted into simulated nonexceedance probabilities and *the volumes are* rescaled using an independently estimated flow-duration curve (FDC) *derived from regional information.*" We will correct this impression throughout the manuscript.
- 40 **Final Author Response 3:** At several places in the manuscript, we have made it clearer that we intend the use of observed FDCs to be an idealized analysis, not an application. For examples, see page 3, line 15, page 8, line 7, and page 9, line 15.

**Reviewer Comment 4:** It is also not clear that the authors want to estimate unbiased time series of discharge at ungauged sites. This has to be clearly stated. One thus needs to have a regional simulation model and a regional model for the FDC estimation. The interest of the configuration where an observed FDC is used to bias correct the simulated time series has to

45 be made clear in this context. This approach is presented as an alternative estimation approach. It is actually not and this is really confusing.

**Author Response 4:** The primary goal of this work is to estimate distributionally unbiased time series of discharge at ungauged sites. This statement will be reinforced. The use of observed FDCs, which, of course, are not available at ungauged locations are presented as an "idealized" case (P1, L10). It is presented merely as demonstration of the "theoretical potential of this [general] approach" (P6, L10), providing evidence to support the hypothesis that such an approach would work with a suitably-estimated FDC (P6, L30). We will strive to make this point more clearly.

5 FDC (P6, L30). We will strive to make this point more clearly. Final Author Response 4: In addressing comments 2 and 3, this comment has been addressed. We included a more explicit statement of the novelty of this work is discussed on page 2, line 3, and on page 6, line 24. Furthermore, we have highlight the idealized utility of the application with observed FDCs (page 3, line 15, page 8, line 7, and page 9, line 15).

Reviewer Comment 5: The results of the study were rather expected. The authors find that the combined "time series simulation"+"FDC bias correction approach" is unable to correct the bias in the simulated hydrograph. The authors explain this is due to the bad quality of the regionalized FDC which present a rather significant distributional bias for all discharge percentiles. By construction, the "poor" result of the FDC bias correction was thus expected. The interest of the method would have been for a configuration where the distributional bias obtained from the "time series simulation" approach is larger than that of the regionalized FDC. Obviously, this is unfortunately here not the case. At least in average for the 1168 basins

15 considered in the work. More convincing results could have perhaps been presented considering different configurations of basins with different relative performances of the two approaches considered here for the combination. We would have seen that the method works in configurations where reginalised FDC are good.

Author Response 5: It is surely expected that this two-step approach will depend on the accuracy of both steps. This project demonstrates the impact that FDC accuracy can have on the results. However, the results were not uniformly poor. As men-

- 20 tioned in the discussion, improvements were seen in certain parts of the distribution of daily streamflow. It is unfortunate that the results were not uniform improvements. In the revision we will identify a few sites that show improvements under the situation the reviewer describes (when bias obtained from the "time series simulation" approach is larger than that of the regionalized FDC). This will supplement our analysis of how performance of the estimated FDC corresponds to improvements in the time series.
- **Final Author Response 5:** In the end, we felt that the addition of figures showing a range of representative performance was not efficient. Instead, we use Figure 10 presented on page 13, line 27, to highlight the variability of performance and augment the discussion in the previous paragraph.

**Reviewer Comment 6:** P2. "As attested to by many researchers focused on the reproduction of historical streamflow, this bias commonly appears as a general overestimation of low streamflow and underestimation of high streamflow". This statement

- 30 is not true. Simulations with any hydrological model will lead to the overestimation of some high stream flow and the underestimation of the others. The same for low flows : some are over-, some are underestimated. In rainfall-runoff models, it clearly depends on the amount of rainfall estimated from the small number of raingauge available, and frequently not numerous enough to have the right estimate of rainfall input. It also depends on the quality of the model which is limited for the simulation of specific hydro processes. The fact that a given model systematically overestimate lowflows and underestimate
- 35 highflows surely means that the model is not good enough and could likely be improved. As a matter of fact, some models present those limitations and this justify the present work. But a general statement can not be given there on such a case. (Note that this statement is invalidated in your work by the results and comments you mention (p6 line 20/25). Please reformulate. Author Response 6: The citations provided do note the tendency of the bias described. Furthermore, this statement is not made categorically. Instead, what is described is a *common* manifestation of bias, not a *categorical* or *general* statement of all bias.
- 40 The results presented do not invalidate this statement, as the medians clearly demonstrate, but do show a range of performance. We will try to make this nuance more clear in the revision of this manuscript.
  Final Author Response 6: We have added an explicit caveat that our claim is not of a universal truth, but rather of a tendency (page 2, line 16). We have also expanded our discussion of the variability of performance starting on page 13, line 27.

**Reviewer Comment 7:* P2. Ln 22. The difference / novelty of the present methodology with that of Fennessey (1994) and Hughes and Smakhtin (1996) is not clear. Please clarify.**

Author Response 7: We will attempt to clarify this in the revision. Fennessey (1994) and Hughes and Smakhtin (1996) presented a method to simulate streamflows using a donor gauge for timing and a regionalized regression for FDCs. The

approach presented in the current work is an extension where the timing is generated from a process-based model (rather than a donor).

Final Author Response 7: The question of novelty is further articulated on page 6, line 24.

Reviewer Comment 8: P3. The description of the content of the paper is missing at the end of the introduction.

- 5 Author Response 8: We will add a paragraph describing the headers of each section as a signpost to the reader.
- Final Author Response 8: This paragraph has been added on page 3, line 11.

**Reviewer Comment 9:** P3. Simulated discharges.  $\hat{A}$  'n initial predictions of daily streamflow values for each streamgauge were obtained by applying the pooled ordinary kriging approach (Farmer, 2016) to each 2-digit Hydrologic Unit". The approach has to be better explained. It is not possible to understand how time series are obtained for any given gauge from what is said in the

- 10 paragraph ("It builds a single, time-invariant semivariogram model of cross-correlation that is then used to estimate ungauged streamflow as a weighted summation of all contemporary observations"). A summary methodological scheme could be helpful. Is the variogram model estimated independently for each day ? for each target location from stations in its neighborhood (or do you use all stations of the United States to estimate the daily discharge of a given station) ? What is a 2-digit Hydrologic Unit ? Does the leave-one out procedure applies for the target location ?
- 15 Author Response 9: Because the underlying hydrologic model is not the novelty of this method nor is it the only simulation method that can be bias-corrected in this way, we chose not to go into detail on the exact simulation methodology. The approach thoroughly described in Farmer (2016) is followed identically here. The semivariance was calculated for all contemporary pairs of daily streamflow observations, after transformation by dividing by drainage area and taking the logarithm, in a 2-digit HUC. These seminvariances were then summarized with a single seminvariogram cloud, which was used to fit a semivariogram model
- 20 that applies equally for all days. The 2-digit Hydrologic Units are the large regions used to classify basins in the United States and shown in Figure 1 (Seaber, Paul R., F. Paul Kapanos, and George L. Knapp (1987). "Hydrologic Unit Maps". United States Geological Survey Water-supply Papers. No. 2294: i–iii, 1–63.). These clarifications will be provided in the revision of this manuscript and we will explore the feasibility of a graphic to summarize the simulation routine.
- Final Author Response 9: On page 4, line 25, the reader is directed to find further discussion of the simulation method
  elsewhere. The motivation for not including these details here, namely that they are published elsewhere and do not represent the novelty of this work, is also highlighted on page 3, line 22.

**Reviewer Comment 10:** P4. Ln 1-8 : Please clarify. "Daily FDCs were developed independently of the streamflow simulation procedure by following a regionalization procedure similar to that of Farmer et al. (2014). You mention "These same percentiles were then estimated using a leave-one-out cross-validation of regional regression". The objective/process of this estimation was not clear. Please make clear that the FDC used to bias correct your target station is obtained from a regional FDC model,

obtained from all (or a part of) the stations close to this target station.

30

Author Response 10: We will provide clarification in the revision. Farmer et al. (2014) use unsupervised regional regression that relies on a best-subsets regression to estimate the complete FDC in a way that tries to capture the dependence between quantiles. The leave-one-out procedure is used to quantify performance as if the target site were completely ungauged. Region-

35 alized regression relies on all stations within a pre-defined region (in this case, the 2-digit Hydrologic Unit).

Final Author Response 10: This clarification is made on page 5, lines 1 and 5.

**Reviewer Comment 11:* P4. Explanatory variables. Please comment the possible correlations between those. Have you looked / used for uncorrelated sets of explanatory variables ?**

Author Response 11: Farmer et al. (2014) discuss how explanatory correlations were controlled. Most of this detail, being
already published elsewhere, is beyond the scope of this current work. However, we also used a limitation of variance inflation factors of each model. This will be added to the revision of the manuscript.

Final Author Response 11: A discussion of explanatory variables has been added on page 5, line 14.

**Reviewer Comment 12: P4 – In 10 : what are "best-subsets regression". Best sub-sets of what ?**

**Author Response 12:** Best-subsets regression is a common regression technique that exhaustively searches the predictor space for the best model with a specified number of variables. The specified number of variables is then changed to explore a range of model sizes. As described by Farmer et al. (2014), of these models that then differ in size, the AIC (or some other metric) is used to select the "best" model in an unsupervised fashion.

Final Author Response 12: A better description of best-subsets regression has been provided on page 5, line 5.

**Reviewer Comment 13:** What is the difference with "three contiguous streamflow regimes you mention in line 17.

5 Author Response 13: The three contiguous regions allow for different explanatory variables to be used to predict different streamflow quantiles. Each is continguous in that it mast span a contained range of quantiles, e.g. (0.02%, 0.05%, 0.1%, etc.) and not (0.02%, 75%, 99.98%). Because three are allowed, these can be thought of as "low", "medium" and "high" streamflows. We will provide this example in the revision.

Final Author Response 13: The discussion of these regimes has been improved with additions on page 5, line 20.

10 **Reviewer Comment 14:** P4; Please clarify. "the percentiles were grouped into a maximum of three contiguous streamflow regimes based on the behavior of the unit FDCs in the region". What is a unit FDC ? How are defined the regimes ? Are they relative to different parts of the FDCs or to different sets of different basins within the region ? Do you group 95th and 90th percentiles for instance ?

Author Response 14: The unit FDC is the duration curve of streamflow divided by drainage area; this definition will be

15 provided in the revision. Based on a national analysis of unit FDCs (Over et al., in press [expected to be published before revision]), it is possible to identify low, medium and high streamflow regimes. This explanation will be provided. Final Author Response 14: The discussion of these regimes has been improved with additions on page 5, line 20.

**Reviewer Comment 15:** P4. "The percentiles in each regime were estimated by the same explanatory variables, allowing only the fitted coefficients to change.." Do you mean that a regression is estimated for each of the Twenty-seven percentiles you

20 considered ? If yes, a very significant risk on a non monotonic behavior of the FDC is likely. Why not working with a analytical model fitted to the FDC (and then propose a regionalization of the parameters of this model ?). This issue should be discussed somewhere.

Author Response 15: The answer to the reviewers question depends on how the reviewer is defining "regression". Within a given regime, as defined earlier, all of the quantiles have the same explanatory variables in the final equation. However, the

- 25 coefficients on those variables are fit independently across quantiles. Because the same variables are used, monotonicity is made much more likely within the regime; the only nonmonotonicity therefore occurs at the separation of regimes. We will add a discussion of this point. Furthermore, identifying the optimal method for FDC prediction is beyond the scope of this work. For this reason, we did not explore analytical solutions for the FDC (an aside: https://doi.org/10.5194/hess-21-3093-2017 found it is very difficult to find a suitable analytical solution).
- **Final Author Response 15:** On page 5, line 32, we have added a discussion of the fitting of coefficients and potential issues of non-monotonicity.

**Reviewer Comment 16:** P4. "Further details on the approach of percentile grouping this methodology can be explored in the associated data and model archive Farmer et al. (2018)." The given reference is a technical report. The soundness of the ""percentile grouping approach" is thus uncertain. It has thus to be fully explained in the present manuscript or it has to be

35 *described in another a peer reviewed journal.*

Author Response 16: The report that the reviewer is looking for is in press and a citation will be added (Over et al., in press [expected to be published before revision). The data release includes all the development code, allowing a user to explore the method in depth, if interested.

Final Author Response 16: The technical report allows the reader to explore the methods further through experimentation. As
they are not the focus nor the foundation of this work, they are not explored here. A publication exploring this methodology is currently in press and expected to be released before publication. Its complete citation has been included in the revision.

**Reviewer Comment 17:** P4. The section on the bias correction approach has to be made clear. Especially how the bias correction works and with which data in the case of ungauged catchments. Again a synthetic graphical scheme of the BC approach would be usefull. You say "The nonexceedance probabilities were then converted to standard normal quantiles

and linearly interpolated along two types of independently estimated FDCs: the regionally regressed FDCs and the observed FDCs determined by applying the Weibull plotting position." It is something to do with quantile mapping? I can imagine that a correction function can be established for any given station where you have both observations and simulations. Is the idea here to transfer the correction function obtained for a gauged catchment to neighboring ungauged ones? If yes, how do you

5 define the neighborhood ? how do you consider the different corrections functions you can obtained for the different gauges stations you may have in the neighborhood of your target location ?

Author Response 17: We will clarify in the revised manuscript that the idealized method that uses observed FDCs is only an example to demonstrate theoretical utility. In practice, only an estimated FDC can be used. There is no development or transferal of correction functions. We will provide a graphical demonstration of the methodology.

10 **Final Author Response 17:** On page 6, line 17, we have improved the discussion of the methods by using a figure. On line 9, we clear up some of the confusion with respect to observed FDCs.

**Reviewer Comment 18:** P4. Ln - 29/30: "The nonexceedance probabilities were then converted to standard normal quantiles and linearly interpolated along two types of independently estimated FDCs: the regionally regressed FDCs and the observed FDCs determined by applying the Weibull plotting position." » This is confusing. In the ungauged basin configuration : only one

- 15 FDC is expected to be used : the regionalized one. The observed FDC is not expected to be available in a ungauged catchment. You use it here only to estimate the added value of the quality of the regionalized FDC on the resulting bias corrected discarhge time series. The "independently estimated FDC" you mention in line 31 (and basically everywhere else in your manuscript) should first refer, I guess, to the regionalized FDC (and not to the FDC from observations).
- Author Response 18: The reviewer is correct, as we have addressed in other comments. The use of an observed case is merely
   for the purpose of demonstration. The reason we differentiate between "independently estimated FDC" and "regionalized FDC" is that regionalization is not the only way to estimate an ungaged FDC. This will be clarified.

Final Author Response 18: This clarification has been made on page 6, line 8, and elsewhere.

25

30

40

**Reviewer Comment 19:** P5. Ln 3: reformulate : "by correcting the simulated volumes to an independently estimated FDC. **Author Response 19:** We will revise to "by *rescaling* the simulated volumes to an independently estimated FDC." **Final Author Response 19:** This change appears on page 6, line 15.

**Reviewer Comment 20:** P5. Evaluation : Ln 10- the two different evaluations approaches were not clear for me at first. A reformulation would be worth Ln 20 & follow. Evaluation for the whole FDC, or for a given tail : what is the evaluation crietiron : the mean of the bias estimated between each pair of percentiles ? the biais between the mean of the percentiles for the raw and corrected data respectively ?? Evaluation on observation-independent tails and observation dependant tails. A graphical scheme please to explain what is done in the second case, at least in a supplementary material !

- Author Response 20: We will provide a graphical scheme of the two types of tails in the revision. Several criterion are provided for evaluation (e.g., average differences in common logarithms, RMSE, NSE). The median across all sites is taken as the average performance for our data set. There is, of course, some spread around the central tendency, and this is discussed, but using the central tendency is a traditional means of assessing bias.
- **Final Author Response 20:** Upon reconsideration, the addition of a figure showing the two types of tails was not an efficient use of space. We have revised the evaluation section on page 6 and believe the techniques for evaluation have been made more understandable.

**Reviewer Comment 21:** Results : all results are given in the form of boxplots. This is likely not enough to understand how the methodology work and how good it is. To give for a selection of stations the different curves (observed / regressed / regressed+bias corrected) would be helpful (with a good performance station and a bad one for instance)

Author Response 21: We will provide a figure of some example cases in the revision.

**Final Author Response 21:** Given there are over 1000 sites considered, the addition of figures exploring the range of example cases was not an efficient use of space. Instead, the figure of methodology (2), a map of original bias (figure 7) and the correspondence of tail bias (figure 10) provide the context the reviewer is looking for.

**Reviewer Comment 22:** P5 ln33 and p7. Ln 25. I find the term "timing" and "error in timing" not appropriate. You could perhaps say "an error in timing of the percentiles". This is however more an error in the temporal structure of your simulated time series. This results from an error in volume which is one day an over-estimation of the true volume and the day after an under-estimation.

5 Author Response 22: We will consider how to clarify this. "An error in timing of the percentiles" is misleading, as it is not the percentiles that provide timing. Timing is provided by the relative ranks and nonexceedance probabilities. By "timing error", we intend to refer to the "error in the temporal structure". We will use this formulation in the revision. It is true that a transposition of the volumes could appear as a timing error, but this is not explored here (a discussion point will be added).

**Final Author Response 22:** In several places, we have removed this reference to timing. Examples are on page 1, line 6 and pages 7, 10 and 11.

**Reviewer Comment 23:** P7 In34. You say "These timing errors also almost result in errors in a particular direction: low for high flow and high for low flows". You have perhaps such a mean behavior but as mentioned above, you have a number of low flows that are overestimated but also a number of low flows (less frequent) that are underestimated. . . Please reformulate to put this statement in perspective.

15 Author Response 23: We will clarify that we are not making a general statement, but observing a central tendency amid a range of values.

Final Author Response 23: This clarification was made on page 10, lines 24 and 26.

25

**Reviewer Comment 24:** P6. Ln 2 : "Figure 4 and Table 1 summarize the tail bias in all approaches to streamflow simulation considered here.". What are the 3 different approaches ? This has to be clearly explained previously. The BC-Obs is not an "approach" similar to the 2 others as you do not know in principle the observed discharge for the target ungauged hasin. It

20 "approach" similar to the 2 others as you do not know in principle the observed discharge for the target ungauged basin. It just allows you to identify the influence of the quality of the regionalised FDC. A reformulation is required when relevant in all the manuscript. The presentation of the method has probably also to be restructered to make it clearer.

**Author Response 24:** This was addressed in a previous comment. A graphical presentation will be provided to distinguish between observation-dependent and observation-independent tails. The BC-Obs is provided for exactly the reason the reviewer describes: not as a viable alternative for application.

**Final Author Response 24:** As with comment 3, at several places in the manuscript, we have made it clearer that we intend the use of observed FDCs to be an idealized analysis, not an application. For examples, see page 3, line 15, page 8, line 7, and page 9, line 15. With this improvement, a graphical representation of observation-dependent and observation-independent tails was not deemed efficient.

- 30 **Reviewer Comment 25:** P7. 1. The analysis of the second paragraph in this section 3.1 is clumsy. The Observed FDC is in principle perfect and thus the bias in simulation for the observation independent evaluation case should fully vanish after correction. You should have nothing or roughly nothing to comment here. The results should be perfect. Why is there some remaining bias with the BC-obs approach ?? Please comment. (could it be a difference in the time period used for the simulation and the time period used to identify the observed FDC? Is it something related to the epsilon value you add to discharge data
- 35 for the logarithmic transformation issue ? to the reduced number of percentile used to describe the FDC ? something else ? ) The comment on this in the conclusion section has also to be modified accordingly (p12 – ln 1/10). P7. 24: "To understand the effect of errors in timing further, consider Figure 6, which shows the mean error in the nonexceedance probabilities of the observation-dependent upper and lower tails." I can not understand (just guess) what is refered to here ? Please clarify. Author Desenance 25: The use of an enserved EDC will still rely on the timing of the circulated hudeograph. It is for this
- Author Response 25: The use of an observed FDC will still rely on the timing of the simulated hydrograph. It is for this
  reason that you are likely to see some residual bias in the observation-dependent tails (Fig. 4, boxplots C and F). With the observation-independent tail (Fig. 4, boxplots I and L), the much smaller residual bias is likely a result of the interpolation along an FDC defined by a finite set of quantiles. There may be some effect from the epsilon value and censored regression used for zero-valued streamflows, but we did not find a major impact. We will add a note on this discussion.
  Final Author Response 25: This discussion was added on page 10, line 18. The figure is now Figure 8.
- 45 *Reviewer Comment 26:* Fig 2 and following : how many data are used for each boxplot (no. stations x no. percentiles ???). Is there one point for each station/percentile ?

Author Response 26: There are 1168 points in each boxplot (one for each station); we will make this clearer in the caption. The metric is the mean difference in common logarithms of the complete FDC at a given site.

**Final Author Response 26:** The captions clearly define the number of sites being presented. The revisions of the evaluation section help to better understand the performance metrics used here.

- 5 **Reviewer Comment 27:** Discussion : what about the likely seasonal dependence of the correction function ? Is there some potential for improvement here ? The estimated FDCs are composed of 27 quantiles, of which the upper and lower tails contain only the eight values with nonexceedance probabilities 95% and larger and 5% and smaller, respectively.). A comment on the number of quantiles used to describe the FDC would be worth (a sensitivity analysis of the results to this number could be also included in a discussion section)
- 10 Author Response 27: No correction function, as such, was used. (It could be conceived as a function, but that is not explored or proposed here.) There is, of course, always room for improvement; we provide a discussion of the sensitivity to the representation of the FDC. We will strengthen that discussion by noting the limitations of an FDC defined by a finite number of quantiles.

Final Author Response 27: This discussion was added on page 13, line 9.

- 15 Reviewer Comment 28: What is the influence of the duration of the observation time period used to estimate the observed FDC > on the quality of the FDC estimation and then on the quality of the bias correction ?
   Author Response 28: As the period of analysis is fixed here, it is not possible to explore this effect. We will make a recommendation for future work, as it will certainly have an impact on something like future projections.
   Final Author Response 28: This addition was made on page 13, line 9.
- 20 *Reviewer Comment 29:* Is there any dependence of the results to the hydroclimatic context of the basin ? How is it structured is space in US ?

Author Response 29: We could not see any obvious pattern: "Initial exploration did not find a strong regional component to performance of the bias correction method. For some regions, like New England, where FDCs are well estimated by regional regression, there is a general improvement in accuracy under bias correction with regionally regressed FDCs, but the improve-

25 ment is highly variable. Instead, the strongest link is with the reproduction of the FDC. " It may be that the hydroclimate is driving the ability to reproduce the FDC through regression, but that is left for future research. Final Author Response 29: We have strengthened this discussion on page 13, line 13 and on.

**Reviewer Comment 30:** Minor remark : P2. Ln10. The interest of the "long term forecast term" here is not clear. It seems to be out of the scope of the work. To be better explained. What is long term ?

30 Author Response 30: Exploration of long-term forecast is beyond the scope of this work and no effort is made to explore it here. Long-term (decadal and beyond) forecasts are mentioned as an example of a hazard of underlying bias. We will clarify this statement by defining long-term as decades and beyond.

Final Author Response 30: We added this specification on page 2, line 14.

*Reviewer Comment 31:* P2. 15/20 : » this paragraph is not clear »> to be clarified / double-formulate. The "interpolation of non-exceedance probabilities along the FDC" is a rather clumsy formulation. What does it mean ?

Author Response 31: Described further in the methodology, the interpolation must occur because the FDC is being represented as a finite set of quantiles (27). If a no exceedance probability does not fall exactly on one of those percentiles (P4, L10), it must be interpolated. We will reference the methodology in the revision.

Final Author Response 31: The reader has been directed to the discussion of methods on page 2, line 27.

**40 *Reviewer Comment 32:* p2. Ln 33 : please clarify what is meant there**

**Author Response 32:** As before, if the nonexceedance probability falls within the range of the quantiles (P4, L10), it can be interpolated. However, it is falls outside of the range of quantiles (P4, L10), it must be extrapolated. The two nearest points were used for linear extrapolation. We will consider how to clarify.

**Final Author Response 32:** Our original response unintentionally missed the mark here. By moving the paragraph referenced by the reviewer to the discussion, the relationship with previous methods can be more easily understood.

**Reviewer Comment 33:** P2. Ln 35 and following. This does not belong to the introduction but to a discussion section. The discussion should probably give the evaluation of the other method suggested here (Additional research to explore if estimating

5 nonexceedance probabilities directly, as opposed to the conversion of simulated streamflow to nonexceedance probabilities used here, might further improve nonlinear spatial interpolation using FDCs or simulation more generally
 Author Response 33: This will be moved to the discussion. Further evaluation is outside the scope of the present work.
 Final Author Response 33: This was moved to the discussion on page 14, line 31.

Reviewer Comment 34: p3. Ln 3 : give the structure of the paper

10 Author Response 34: This will be added, recapitulating the headers of the document. Final Author Response 34: This was added on page 3, line 11.

**Reviewer Comment 35:** p5. "The root-mean-squared error of the common logarithms of streamflow and the differences therein were used to quantify accuracy." Do you mix streamflow and differences between streamflow in the computation of a single RMSE criterion ? If yes, I fear it is not relevant or please clarify / justify.

- 15 Author Response 35: We do not mix streamflow and differences between streamflow in the computation of a single RMSE. The RMSE is calculated for two different approaches; we then observed the differences across approaches. We agree that this setnence is confusing, so will explore other options. One possible revision could be "The differences in the root-mean-squared error of the common logarithms of the predicted streamflow for the two approaches were used to quantify accuracy." Final Author Response 35: On page 5, line 1, we have added formulas to help the reader understand the computations.
- 20 Reviewer Comment 36: P6. Ln 9 add a subsection title "simulated hydrographs without correction" Author Response 36: This section will be added.
   Final Author Response 36: This has been added on page 8, line 21.

*Reviewer Comment 37:* P6 - 25. "These results show upward bias in lower tails and downward bias in upper tails." No, this is not the case in general. See your paragraph above.

Author Response 37: We do not claim that the result is general, but the result do show a tendency in the direction described. We also discuss the variability of this performance, but this will be further highlighted in the revision.
 Final Author Response 37: On page 9, line 10, we explicitly state that we are making observations on central tendency and not claiming universality.

*Reviewer Comment 38:* P9 "For the observation-independent case, the errors are removed almost completely, and the remaining errors in the observation-dependent case mimic the timing (nonexceedance probability) errors." This is not true (only if the observed FDC is used)

Author Response 38: The first half of this paragraph only discusses the case where observed FDCs are used. We will clarify this transition.

Final Author Response 38: Clarification was added on page 12, line 13.

35 *Reviewer Comment 39:* P10. 2 : "The change in the absolute bias of the observation-independent lower tail has a 0.72 Pearson correlation with the absolute bias of the lowest eight percentiles of the FDC estimated with regional "regression." I do not understand what is meant here.

Author Response 39: We will add, "That is, the residual bias in the bias-corrected FDC is strongly correlated with the bias in the independently-estimated FDC."

40 Final Author Response 39: This has been added on page 13, line 5.

**Reviewer Comment 40:** Figures P10- 6 "as regional regression is not the only tool for estimating FDCs, improved methods for DC estimation would further increase the impact of this bias correction procedure." Mention other such methods.

Author Response 40: Moving and expanding the last paragraph of the introduction will allow us to discuss other methods. Some might include TNDTK, kriging methodologies, index-flood methods, other hydrograph simulations, etc. Final Author Response 40: This was added on page 6, line 3; page 13, line 9; and page 15, line 1.

Reviewer Comment 41: P10.15 "It may not always be possible to determine the accuracy with which a given FDC estimation

- 5 technique might perform, making it difficult to determine if these results can be generalized." There is no reason why the accuracy of any FDC regionalization approach could not be assessed (this is done in all FDC regionalization study). At least with a leave-one out procedure. Before applied for the bias correction of any time series simulation model, this quality should be checked and estimated to be better than that of the simulation model.
- Author Response 41: We will revise to, "*At a particular ungaged location*, it may not always be possible to determine the accuracy with which a given FDC estimation technique might perform (*beyond a regional cross-validation assessment*), making it difficult to determine if these results can be generalized."

Final Author Response 41: The revision can be seen on page 13, line 23.

**Reviewer Comment 42:** Fig. 1 – What is meant here : "The outlines of 2-digit Hydrologic Units are provided for further context.

15 Author Response 42: The polygons represent the 2-digit Hydrologic Units, which are the regionalization areas. The meaning of these units is described in an earlier comment and will be incorporated into the revised manuscript. Final Author Response 42: A description of 2-digit hydrologic units has been added on page 4. line 18.

**Reviewer Comment 43: Fig. 2 : BC-RR and BCObs have to be defined in the main text.**

Author Response 43: We will consider adding these into the report. They are not currently used in the main body, as the repetition of opaque acronyms might detract from clarity.

**Final Author Response 43:** We considered using these in the report, but felt that remove some degree of clarity. The acronyms are defined in teh figure captions alone so that the figures might be understood independently.

**Reviewer Comment 44:** Figures and tables : simplify the captions : a number of repetitions could be removed (and a reference to the caption details of one reference figure added in the caption of all other figures)

25 Author Response 44: We prefer that the captions allow each figure to standalone as much as possible. Final Author Response 44: We have retained the formulations in the captions to provide the most clarity for each figure independently.

**Reviewer Comment 45: Are the 3 tables usefull ?**

Author Response 45: We feel they are useful, as they provide a summary of the numbers discussed in the report.

30 Final Author Response 45: The tables have been retained.

**3 Third Anonymous Reviewer**

20

**Reviewer Comment 1:** The authors presented a bias-correction procedure useful for improving the accuracy of simulated daily streamflow series by using independently estimated flow-duration curves (FDCs). Although the procedure itself is not completely new (references of previous studies are included in the manuscript), the study is interesting as it considers an

35 extended database in the US and focuses on the reproducibility of upper and lower tails, distinguishing between observationdependent and observationindependent tails. This aspect is meaningful for highlighting the effect of timing on distributional bias. The study concludes that the significant potential of the bias-correction procedure is limited by the accuracy of the FDCs estimation method.

The paper is well structured and written, but some changes should be applied for making it more readable and for empha-

40 sizing some important aspects. I believe it is suitable for publication in HESS after the authors address some issues reported in the comments below.

Author Response 1: Thank you for your deep consideration of our manuscript. Your kind comments will surely help us to improve the delivery of our results.

**Final Author Response 1:** We wish to again extend our thanks. Your comments were very useful in thinking about how to better describe our methods and present our findings. The revised version of our manuscript is certainly much stronger because

5 of your input.

**Reviewer Comment 2:** I would suggest to explain more in details the "Bias correction" procedure at Section 2.3. For instance, at Page 4 Lines 29-30, the sentence "[...] linearly interpolated along two types of independently estimated FDCs" is rather ambiguous: I found difficult to understand whether the authors refer to the resampling of the curves, or perhaps to the prediction of FDC quantiles, which is carried out with a linear regression. I would recommend to rephrase this sentence and add more

- 10 information to it, in order to clarify better this fundamental aspect of the proposed procedure. Moreover, at least one figure could be useful for clarifying the procedure. I can suggest to show at least two plots, where the authors may report standard normal quantiles vs. logarithmically transformed streamflow percentiles for, in turn, (a) regionally regressed FDC and pooled ordinary kriging curve, and (b) observed FDC and pooled ordinary kriging curve. Finally, in my opinion, the bias-correction section should be extended: I recommend to add more detailed information about how the bias correction is applied to the simulated streamflows, maybe introducing a figure vignette, or, likewise, describing the procedure point by point.
- 15 simulated streamflows, maybe introducing a figure vignette, or, likewise, describing the procedure point by point. Author Response 2: In concert with the response of other reviewers, we will be reformatting the section on methodology. It will be substantially enhanced by figures that show example hydrographs and example FDCs, as well as a figure demonstrating the steps of the procedure.

Final Author Response 2: The section on methodology has been revised. In particular, page 6, line 17, of the tracked-changes
document presents a figure more completely describing the methodology for bias correction.

**Reviewer Comment 3:** I would stress more that in the majority of possible practical applications of the proposed method (i.e. predictions in ungauged sites), using observed FDCs for the bias correction would not be possible. Indeed, the only exception could be represented by those catchments in which we want to simulate streamflows for a given period, even though we have streamflow data for another period. I would add this reasonings to the revised version of the paper.

25 **Author Response 3:** We will add clarification to point out that the use of observed FDCs is only provided as a theoretical demonstration of the upper-limit of performance in ungauged locations. We did not consider use of observed FDCs for record extension, but this approach might be useful in partially gauged locations. Though we did not explore that particular application, we will mention it in the discussion.

Final Author Response 3: In several places we have highlighted the theoretical utility of observed FDCs, noting that they
are not intended for application (page 3, line 15; page 8, line 7; and page 9, line 15.). Furthermore, we have acknowledge the potential for record extension on page 8, line 7.

**Reviewer Comment 4:** [3] Page 10, Lines 11-13 – The authors highlight that "initial exploration did not find a strong regional component to performance of the bias correction method". In order to better support this sentence and to improve the effectiveness and completeness of the study, the authors could better discuss the spatial distribution of performance, especially

- 35 given the high climatic variability among the conterminous United States. Therefore, I would suggest to add and discuss a new figure (or figures; e.g. a set of maps, similar to Figure 1), showing the spatial distribution of bias and root mean squared error in the study region for at least a couple of the cases considered in the study (Orig., BC-RR, BC-Obs.; upper tail, lower tail; observation-dependent tail, observation-independent tail; sequential, distributional, etc.).
  Author Response 4: We will develop such a figure.
- 40 **Final Author Response 4:** Figure 7 contains a map of the original bias. Figure 10 shows how this bias in each tail might change under bias correction. Maps were not produced for all cases, as they would have been largely redundant with these additional figures.

*Reviewer Comment 5:* In Figure 1, the differences between not-selected and selected streamgauges are not clear: in some areas, crosses overlap with points and the distinction is not simple. Maybe using different colors and symbol sizes might
highlight better the differences between these two categories.

Author Response 5: We will work with the journal to improve the visibility of this figure. Final Author Response 5: We have changed the symbols to showed only included sites, but will continue to consult with the journal.

Reviewer Comment 6: Page 3, Line 24 – It is not clear what a 2-digit Hydrologic Unit is. Could you please explain?

- 5 Author Response 6: The Hydrologic Unit system is a common method for delineating watersheds in the US. 2-digit hydrologic units (the polygons in Figure 1) roughly align with the major river basins of the United States. We will add this description with appropriate citations. In the figure, the units are the outlined polygons. (Seaber, Paul R., F. Paul Kapanos, and George L. Knapp (1987). "Hydrologic Unit Maps". United States Geological Survey Water-supply Papers. No. 2294: i–iii, 1–63.) Final Author Response 6: Hydrologic Units are defined on page 4, line 18.
- 10 **Reviewer Comment 7:** Page 3, Line 16 (and elsewhere) "cubic feet per seconds (cfs)" is used. I believe that the International System of Units should be used in scientific papers. At the same time, if cfs is the standard adopted by USGS (and in the GAGES-II database), I think that at least the conversion factor to m3 s-1 should be indicated in parentheses the first time that cfs is mentioned.

Author Response 7: We will add a conversion factor.

15 Final Author Response 7: The conversion factor was added on page 4, line 8.

**Reviewer Comment 8:** Page 4, Line 3 - It seems that you are referring to period-of-record FDCs; am I correct? Could you please state that explicitly?

Author Response 8: Yes, you are correct. We will state this more clearly. Final Author Response 8: This was added on page 2, line 26, and page 4, line 29.

20 *Reviewer Comment 9:* Page 4, Line 10 - It is not clear to me what "best-subsets regression" is. Could you please clarify and/or add at least one reference?

**Author Response 9:** Best-subsets regression is a common regression technique that exhaustively searches the predictor space for the best model with a specified number of variables. The specified number of variables is then changed to explore a range of model sizes. As described by Farmer et al. (2014), of these models that then differ in size, the AIC (or some other metric) is used to select the "best" model in an unsupervised fashion.

used to select the "best" model in an unsupervised fashion.Final Author Response 9: Best-subsets regression is further defined on page 5, line 5.

*Reviewer Comment 10:* Page 4, Lines 13-15 and 20-22; Page 8, Lines 19-21 - I would suggest to remove parentheses. Author Response 10: We will remove them. Final Author Response 10: These were removed on page 5, line 11, 14, 26 and 28; and page 11, lines 15 and 16.

30 *Reviewer Comment 11:* Page 5, Lines 9-10 – I would recommend to add an equation showing the expression "ten to the power of the difference and subtracting one from this quantity". I would also suggest to explain why you are referring to this equation for computing the percentage. Moreover, the authors could show equations for bias and root mean squared error, respectively, when introducing them.

Author Response 11: We will add these equations and any relevant citations.

35 Final Author Response 11: These equations were added on page 7, line 1.

**Reviewer Comment 12:** Page 5, Line 10 – The authors write "root-mean-squared error", while use "root mean squared error" in the remainder of the text. I would suggest to use the same expression everywhere ("root mean squared error" should be fine). Author Response 12: We will make every effort to ensure consistency throughout the manuscript. Final Author Response 12: This was changed on page 7, line 8.

 40 Reviewer Comment 13: Page 5, Line 13 – Could you please add a reference for the Wilcoxon signed rank test? Author Response 13: We will add this citation: Wilcoxon, Frank (Dec 1945). "Individual comparisons by ranking methods". Biometrics Bulletin. 1 (6): 80–83. doi:10.2307/3001968. Final Author Response 13: This citation was added on page 7, line 13.

**Reviewer Comment 14:** In Tables' captions, the acronyms "OD" and "OI" are used instead of "observation-dependent" and "observation-independent", respectively. You could use these acronyms also in the body of the text, in order to improve the readability; e.g. "OD-tail" instead of "observation-dependent tail".

5 Author Response 14: A previous reviewer suggested also using BC-Obs, etc. We will attempt to use these abbreviations, and ensure that clarity is not harmed.

**Final Author Response 14:** After using these abbreviations, we felt they removed a bit of clarity from the prose. We decided not to use abbreviations and keep the explicit descriptions.

**Reviewer Comment 15:** Page 10, Lines 6-7 – Could you please report some other methods (please provide references) for estimating FDCs?

Author Response 15: Moving and expanding the last paragraph of the introduction will allow us to discuss other methods. Some might include TNDTK, kriging methodologies, index-flood methods, other hydrograph simulations, etc. Final Author Response 15: This is appended on page 15, line 1.

Reviewer Comment 16: Page 11, Line 16 - Please delete "Summary and conclusions".

15 Author Response 16: Yes, we will fix this error.Final Author Response 16: This has been revised on page 15, line 3.

10

**Reviewer Comment 17:** Captions of Figures 2, 3, 4, 5 – Please remove comma in "pooled, ordinary kriging". Author Response 17: Commas will be removed. Final Author Response 17: Commas were removed in the figure cations.

20 Reviewer Comment 18: In the body of text, there are some references to a recent study by Pugliese et al. (2017), which is currently under review. I would suggest to update these references after its possible acceptance/publication. Author Response 18: We have been keeping an eye on this publication and will update the references when a decision on that manuscript is made.

**Final Author Response 18:** At the time of writing, this work has not been published. We will continue to monitor its progress in hopes that it is published before this work goes to press.

**Bias correction of Simulated Historical Daily Streamflow at Ungauged Locations by Using Independently Estimated Flow-Duration Curves**

William H. Farmer1, Thomas M. Over2, and Julie E. Kiang3

1U.S. Geological Survey, Denver, Colorado, United States
2U.S. Geological Survey, Urbana, Illinois, United States

[revised manuscript text omitted]

---

## Author Response (AR2)

We deeply appreciate the time and effort that the handling editor and the reviewers have put into improving this manuscript. The following contains all comments and our responses. Changes can be seen in the tracked-changes document.

**1   Editor's Comments**

*The Authors have seriously taken in consideration the main comments of the reviews and provided a stronger revised manuscript. Some additional changes (see in particular rev 3) are still requested in order to better convey the main message of the manuscript. In this spirit, I encourage the authors to further review their work. Then I will be in the position to finalize the assessment of the manuscript.*

**Author Response:**

We have considered the comments of all reviewers. The re-reviews by reviewers one and two were quite favorable and helped us to further refine the document. They report that the revised manuscript has effectively improved our ability to communicate our findings and our message.

Reviewer three has provided of a review of the un-revised, original manuscript. Not surprisingly, he or she identified many of the concerns of previous reviewers. Of course, these were addressed in the revised manuscript that the reviewer did not consider. In correspondence with the handling editor, we were advised to consider the spirit of the comments and make any additional changes that were not addressed by previous revision. We have done so.

The revised manuscript, included here with tracked changes, shows our minor revision to the manuscript guided by the comments of all three reviewers. We thank the journal, handling editor and reviewers for their continued consideration of our manuscript and hope they will find this revision satisfactory.

**2   First Reviewer**

**Reviewer Comment 1:** *The authors have significantly improved their manuscript and clarified their work. The methodology is now clear to me and the discussion of results are fair, including the poor performance of the overall bias correction method (as mentioned in the manuscript, rooms for improvement obviously exist with a better estimation of regional Flow Duration Curves). Almost all my previous suggestions / questions have been addressed. The manuscript now just requires some minor revisions. I do not need to review the forthcoming corrected version.*

**Author Response 1:** Thank you for you assessment of this work. Your attention to detail has greatly strengthened the communication of our findings.

**Reviewer Comment 2:** *P1/20 : over the last 20years at least..*
**Author Response 2:** Revised to read "Over the past 20 years, at least, research..."

**Reviewer Comment 3:** *P2/16 : what is a categorical truism. Please give a more accessible formulation.*
**Author Response 3:** Revised to "universal truth that occurs without variation".

**Reviewer Comment 4:** *P2/20 : "rank-order of simulated streamflows"*
**Author Response 4:** Revised as suggested.

**Reviewer Comment 5:** *P2/22 : "the temporal sequence of relative..."*
**Author Response 5:** Revised as suggested.

**Reviewer Comment 6:** *P2/25 : the expression is not clear : "to rescale the streamflow simulations by interpolating the nonexceedance probabilities of the simulated streamflow along the FDC (see Material and Methods, below).*
*Reformulate to : to rescale each streamflow value based on the streamflow value of the regional FDC for the corresponding nonexceedance probabilities (see Material and Methods, below).*
**Author Response 6:** Revised as suggested.

***Reviewer Comment 7:*** *P2/30 : "hydrograph" typically refers to a given hydrological event, not to the time series of discharge for a n-years period. Next the spatial issue is missing in the first part of the sentence.*

*Reformulate to : "The approach tested here seeks to bias-correct a simulated time series of daily discharge using an independently estimated FDC, when viewed in another way, presents a novel form of nonlinear spatial interpolation".*

**Author Response 7:** Revised as suggested.

***Reviewer Comment 8:*** *P3/15 : typo : simulation*
**Author Response 8:** Revised as suggested.

***Reviewer Comment 9:*** *P7/19-20 : "sequential" and "distributional" is used in the different figures but not defined. Introduce both terms here :*

*In the same fashion, evaluation of the complete hydrograph can be assessed sequentially (sequential evaluation in the following), retaining the contemporary sequencing of observations and simulations, or distributionally (distributional evaluation in the following), considering the observations and simulations ranked independently.*
**Author Response 9:** Revised as suggested.

***Reviewer Comment 10:*** *P8/22 : Figure 7 is introduced but not commented > please comment.*
**Author Response 10:** This figures is discussed later. We have added a parenthetical pointer to the later discussion.

***Reviewer Comment 11:*** *P9/35 : You can conclude this. Figure 4 and 6 do not discuss the bias but the accuracy. Please reformulate.*
**Author Response 11:** The first clause refers to Figure 5, while the second clause refers to Figure 6. We have added parenthetical references.

***Reviewer Comment 12:*** *P9/33 : I do not see those numbers (30%, 20%) in the table.*
**Author Response 12:** The table reports raw bias, while the transformation in equation 2 allows us to report bias as a percentage. We have added a parenthetical note.

***Reviewer Comment 13:*** *P10/38 : "from errors in the simulated temporal structure"*
**Author Response 13:** Revised as suggested.

***Reviewer Comment 14:*** *P12/2 : "independent evaluation cases. . . . . . . " and again "independent evaluation case. . . .*
**Author Response 14:** Revised as suggested.

***Reviewer Comment 15:*** *P13/12. I do not understand what you conclude from the figure. How can you state that there is a "general move towards unbiasedness" > I do not think the figure allows such a conclusion (I would even conclude the reverse from graph d) : we should see a compaction in the Y axis but nothing is obvious !!! Perhaps give the limits of the standard deviations (or some high /low percentile) of the cloud in both X and Y directions*
**Author Response 15:** Revised to read: "While there is a move towards unbiasedness at some sites (along the vertical axis), there is a great degree of variability that makes it difficult to draw general conclusions. In some situations, as in Panel D, the variability may actually be increasing with bias correction. Though all methods will produce variability, it remains to future research to determine if a more consistent representation of the FDC might reduce the variability of this performance."

***Reviewer Comment 16:*** *P13/31-32 : I do not understand. Please clarify the "principle" of this alternative approach*
**Author Response 16:** We added the following sentence: "Here, the nonexceedance probabilities were derived from a simulation of the complete hydrograph. In this alternative approach, the discharge volumes would not be estimated but rather only the daily nonexceedance probabilities."

***Reviewer Comment 17:*** *P14 /14-16 : already discussed previously > to be removed.*

**Author Response 17:** Revised as suggested.

*Reviewer Comment 18:* P15/20 : *"While using the nonexceedance probabilities of kriged streamflow simulations 5 improves upon the use of single index streamgages to obtain nonexceedance probabilities,". The use of single index streamgages has not been considered here. You can not say your method "improves...." > refomulate*

**Author Response 18:** Revised to say "may improve"

**3   Second Reviewer**

*Reviewer Comment 1:* *The authors have properly replied to all my comments and I believe that the current version of the paper is more readable and clear than the previous one. In my opinion, the paper is suitable for final publication in HESS after the authors address some minor technical corrections.*

**Author Response 1:** Thank you for your encouragement.

*Reviewer Comment 2:* *Page 4, Line 1 – As the new version of Figure 1 is reporting only the selected streamgauges, I would suggest to remove "and not-selected".*
**Author Response 2:** Revised as suggested.

*Reviewer Comment 3:* *Page 10, Line 2 – "The temporal of the simulated hydrograph [...]". Is the word "structure" missing? In this case, please correct with "The temporal structure of the simulated hydrograph [...]".*
**Author Response 3:** Revised as suggested.

**4   Third Reviewer**

**Author Comment:** This reviewer provided comments on the unrevised, original manuscript and did not consider the manuscript revised after the first round of reviews. As such, several of his or her comments have already been addressed in previous revisions. However, at the direction of the handling editor, we have sought to address the heart of comments rather than focus on the details of the comments. The handling editor asked that we keep these comments on the old manuscript in mind as we produce minor revisions.

*Reviewer Comment 1:* *The paper presents a method to bias correct streamflow by modifying an existing approach of using independent FDC. Although the paper addresses a very interesting topic, the readability of the manuscript and presentation of results are unfortunately very poor. For instance, abstract contains words like '... bias arising from distributional properties of residuals', '... a method for rescaling simulated streamflow to correct...', '... pooled ordinary Kriging simulation', '... regional regression on basin characteristics', '... using an idealized case', 'region of neighbors to ...'. I noticed that these words are mostly coming from Authors' previous papers. However, to make it an independent paper and easily readable, Authors should consider clarifying these points in the abstract.*
**Author Response 1:** Based on previous reviews, the readability of the manuscript has been greatly improved. The detailed explanation of topics raised in the abstract is provided in the remainder of the manuscript.

*Reviewer Comment 2:* *There are many sentences in the Introduction which appear confusing because of the kind of words used or complex way of trying to say too many things in a sentence. For example, '... distributional bias in simulated streamflow is a failure to reproduce....'. Bias is not a failure, it is an error. '... effective squeezing of the streamflow distribution'; what does the word squeezing mean here? '.... distributional compaction...', so it means squeezing lead to compaction; compaction of what? 'This bias is particularly concerning, as examinations of extreme high-flow events are a common and influential use of*

*historical simulation and long-term forecast.' What does this sentence mean? 'Consider, for example, the motivation for work by Archfield et al.' Does it mean that the readers should go through all the cited papers?*

**Author Response 2:** This comment is similar to those addressed in previous revisions. We have made the following additional revisions: (1) We revised the manuscript to read '... distributional bias in simulated streamflow is an error in reproducing....';
(2) We revised the manuscript to read '... effective squeezing of the streamflow distribution, bringing the tails of the distribution closer to the central values'; (3) For consistency, we revised 'distributional compaction' to the synonymous term 'distributional squeezing'. We then go on to discuss how extreme high-flows (e.g., floods) are important hydrologic responses. While it is not unreasonable to imagine that a reader in pursuit of additional information will read cited works, the sentence immediately following the Archfield reference provides the background requested by the reviewer.

**Reviewer Comment 3:** *On page 2, Authors are talking about nonlinear spatial interpolation of daily streamflow using FDCs. The claim is that in previous studies, only single neighbour was used and the nonexceedence probabilities were interpolated along a FDC. It is not clear then how it became nonlinear spatial interpolation when interpolation has to be performed along FDC. All these concepts are presented in few sentences which need further explanation unless the assumption is that the reader will go back and read all the referred papers.*

**Author Response 3:** This is a common description of the approach that has been used by several authors. Interpolation along the FDC is considered nonlinear because it uses a nonlinear FDC. Furthermore, additional tools were introduced in previous work that used standard normal transformations to further highlight the nonlinear nature of the interpolation. The methodology is clearly described with new figures in the revision. It is not unreasonable to expect the readers seeking additional information to pursue cited references, but the information required to understand the methodology is included in this work.

**Reviewer Comment 4:** *'Furthermore, though necessarily explored in this study through the use of a single technique for hydrograph simulation, this approach may be a means to effectively bias-correct any simulation of streamflow, including those from rainfall-runoff models, as presented by Pugliese et al. (2017). Pugliese et al. (2017) used a geostatistical tool...... underlying methods.... the approach proposed by Pugliese et al. is the same as that explored here.' What is the single technique; which geostatistical tool was used; what are the underlying methods mentioned here; etc. need to be explained properly.*

**Author Response 4:** This discussion point is expanded in the revision and in the discussion section. The details of the simulation methodology proposed by Pugliese is already published in his work, and so is not relevant here. As the discussion demonstrates, the relevant connection is the conception of an independently estimated FDC used to bias correct a hydrograph simulation.

**Reviewer Comment 5:** *More information is required to explain how by estimating nonexceedence probabilities directly or indirectly affect the nonlinear spatial interpolation.*

**Author Response 5:** Based on comments from the earlier review, this comment was addressed in the revised manuscript that was overlooked. The discussion of direct and indirect estimation of the nonexceedance probabilities is an interesting point for further research, but not the focus of this work.

**Reviewer Comment 6:** *On page 2 and at the beginning of page3: 'Although the results presented here are promising, they demonstrate that the performance of two stage modeling, where timing and magnitude are largely decoupled, is limited by the less well performing stage of modeling.' This sentence doesn't connect with the previous sentences in the same paragraph. Does it mean that by improving nonlinear spatial interpolation, both timing and magnitude of simulated streamflow are going to improve?*

**Author Response 6:** The previous sentences have been removed per recommendation from other reviewers.

**Reviewer Comment 7:** *On page 3: Please use SI units in the paper. '.... the small additive value applied was 0.0049cfs...... the bias in streamflow plus a correction factor.' It appears that Authors are trying to avoid logarithm of a zero. Why not add this correction to only zero streamflow values? Are you adding 0.0049 cfs to entire data or only zero values?*

**Author Response 7:** The Imperial units are used because that is the original reporting limit. This is relevant because the number of figures in the original unit is what dictated rounding by the collecting agency (USGS). The conversion factor has

been provided. Adding a correction to only zeros would alter the underlying streamflow distribution and is contrary to most literature on censoring. By adding a small value to all streamflows, the distribution, in its entirety, is shifted "to the right" and remains intact.

*Reviewer Comment 8: Section 2 says materials and methods; what does the materials mean here? I recommend separating the description of data and study area in a section, then present another section describing the methodology in detail? The methodology is presented in a summary form assuming that readers will go through Authors' previous papers on this topic. I am sure it will take at least two pages description along with equations and figures to describe what is written in a paragraph citing other papers: 'Though the potential for distributional bias applies to any hydrologic simulation (Farmer and Vogel, 2016), for this study, initial predictions of daily streamflow values for each streamgage were obtained by applying the pooled ordinary kriging approach (Farmer, 2016) to each 2-digit Hydrologic Unit (figure 1) through a leave-one-out cross-validation procedure on the 25 streamgages within the 2-digit Hydrologic Unit. This approach considers all pairs of common-logarithmically transformed unit streamflow (discharge per unit area) at each day and builds a single, time-invariant semivariogram model of cross-correlation that is then used to estimate ungauged streamflow as a weighted summation of all contemporary observations. A spherical semivariogram was used as the underlying model form. Additional information on the time series simulation procedure is provided by Farmer (2016).' There are so many concepts involved here, Authors should try to try to explain the steps systematically with the help of equations and figures.*

**Author Response 8:** Material refers to the data used to run this analysis. This heading is a style common in many journal articles. The first subsection in the revised manuscript provides a summary of the data and study area. Comments from the previous round of reviews provided significant revision to this section and addressed the concerns raised by this reviewer. Equations and figures have been improved to better communicate the methodology. For readers requesting additional information on the methods tangential to the main message (e.g., how hydrographs were originally simulated, which is not the focus of this work), it is not unreasonable to expect them to pursue the cited literature for common, previously-documented methods.

*Reviewer Comment 9: The last sentence on page 3 about the pooled Kriging is very general. It has to be specific whether the methodology used here is applicable to other study area or not.*

**Author Response 9:** The previously-revised manuscript contains a clarification on this point. Namely, the reader is reminded that this work is not intended to explore the applicability of the pooled kriging methodology, but rather to present a method for bias correction of an arbitrary method. Detailed exploration of the pooled kriging approach is available in previous publications.

*Reviewer Comment 10: On page 4, again Authors refer to Farmer et al. (2018) and present all the details of regional regression in a paragraph. Without reading the cited paper, there is no detail available for the reader to figure out the form of regression equation, three streamflow regimes, fitted coefficients, percentile groupings etc.*

**Author Response 10:** As the focus of this work is not the particular estimation method, but rather the bias-correction procedure, further detail is left to the readers to pursue. We are not advocating any particular methodology for the estimation of nonexceedance probabilities or FDCs. In our view, it is not considered overly burdensome to cover previously-published methodologies in a few paragraphs and leave further, nonessential details for the reader to pursue. That said, revisions to the original manuscript, which have been overlooked here, addressed much of this concern. The presentation of regional regression, for example, was heavily revised and clarified.

*Reviewer Comment 11: On page 5, Authors present two methods to discuss about the biases in the tails - observation dependent and observation independent tails. Authors should present some figure to explain the concepts rather than summarizing all the analysis in a paragraph.*

**Author Response 11:** Responses to the previous review clarified this section. As the tails are defined by how one ranks the values, we could not conceive of a useful figure for this demonstration. As acknowledged by the two other reviewers, the revised description explicitly states the differences in these approaches.

*Reviewer Comment 12: Instead of commenting on the rest of the paper, I would like to strongly recommend that Authors should use hydrographs, flow duration curves etc. in presenting results. Explaining all the results using box plots is not very practical and it is hard to evaluate what Authors have presented in Section 3.*

**Author Response 12:** In response to the previous round of reviews, this comment was addressed with addition of Figure 2 (revision). Presenting over 1000 different hydropgraphs and FDCs for independent evaluation is not an efficient use of space, though the accompanying data release allows interested users to explore this work further.

*Reviewer Comment 13: In my opinion, the manuscript needs to be re-written thoroughly. Technical contents and novelty can be evaluated only when all the necessary details are present.*
**Author Response 13:** We regret that the reviewer felt unable to provide any technical review. The revisions, which were inadvertently neglected by the reviewer, represent a complete re-write of the work.

[revised manuscript text omitted]
 | -0.2056 | -0.3405 | 0.3138 | 0.3957 | <0.0001 | -0.3702 | -0.5763 | 0.6399 | 0.6027 | <0.0001 | -0.1450 | -0.2224 | 0.1805 | 0.2852 | <0.0001 |